# Improving Long-Range Interactions in Graph Neural Simulators via Hamiltonian Dynamics

**Tai Hoang**[†, 1]  **Alessandro Trenta**[*, 2]  **Alessio Gravina**[*, 2]
**Niklas Freymuth**[1]  **Philipp Becker**[1]  **Davide Bacciu**[2]  **Gerhard Neumann**[1]

[1]Karlsruhe Institute of Technology  [2]University of Pisa

## Abstract

Learning to simulate complex physical systems from data has emerged as a promising way to overcome the limitations of traditional numerical solvers, which often require prohibitive computational costs for high-fidelity solutions. Recent Graph Neural Simulators (GNSs) accelerate simulations by learning dynamics on graph-structured data, yet often struggle to capture long-range interactions and suffer from error accumulation under autoregressive rollouts. To address these challenges, we propose Information-preserving Graph Neural Simulators (IGNS), a graph-based neural simulator built on the principles of Hamiltonian dynamics. This structure guarantees preservation of information across the graph, while extending to port-Hamiltonian systems allows the model to capture a broader class of dynamics, including non-conservative effects. IGNS further incorporates a warmup phase to initialize global context, geometric encoding to handle irregular meshes, and a multi-step training objective that facilitates PDE matching, where the trajectory produced by integrating the port-Hamiltonian core aligns with the ground-truth trajectory, thereby reducing rollout error. To evaluate these properties systematically, we introduce new benchmarks that target long-range dependencies and challenging external forcing scenarios. Across all tasks, IGNS consistently outperforms state-of-the-art GNSs, achieving higher accuracy and stability under challenging and complex dynamical systems. Our project page: https://thobotics.github.io/neural_pde_matching.

## 1 Introduction

Simulating complex physical systems is central to many scientific domains. These systems are typically governed by partial differential equations (PDEs), which are rarely solvable in closed form. To approximate the solution, classical numerical solvers (Dormand, 1996; Ascher, 2008; Evans, 2010) discretize the domain, obtaining a mesh in the case of the finite element method (Brenner & Scott, 2008). However, most solvers are tied to a specific kind of discretization, and sufficiently accurate simulations require fine-grained meshes, which quickly become prohibitively expensive.

In contrast, neural simulators directly learn system dynamics from data (Guo et al., 2016; Bhatnagar et al., 2019; Li et al., 2019c; Pfaff et al., 2021), enabling simulations that can be several orders of magnitude faster than numerical methods. In particular, Graph Neural Simulators (GNSs) (Pfaff et al., 2021; Linkerhägner et al., 2023; Yu et al., 2024) leverage Graph Neural Networks (GNNs) (Wu et al., 2020; Gravina & Bacciu, 2024) to capture interactions between entities, such as mesh elements, through message passing (Gilmer et al., 2017). By encoding local geometry in graph edges, they achieve accurate and efficient simulations that generalize across meshes in applications ranging from fluid to elastic dynamics (Sanchez-Gonzalez et al., 2020; Pfaff et al., 2021; Yu et al., 2025).

GNSs are commonly trained with a one-step loss to predict the next state from the current observation, and then rolled out autoregressively to generate long trajectories during inference. While being effective over short horizons, even small global errors accumulate rapidly. Implicit (Pfaff et al., 2021; Brandstetter et al., 2022) and explicit (Würth et al., 2025) denoising objectives alleviate this to some extent by reducing local noise, but the loss still fails to capture low-frequency drift that

---

[†]Correspondence to tai.hoang@kit.edu.  [*]Equal contribution.

Figure 1: A high-level overview of the proposed IGNS. The model takes as input the initial node state $\bar{X}$ and performs an $l$-step warmup phase (left), enabling each node to incorporate broader spatial context before the rollout begins. The enriched state is then used to initialize the rollout $X^{(0)} = \bar{X}^{(l)}$ (middle). During the simulation phase (right), the system evolves according to the port-Hamiltonian dynamics of Eq. (6), while the multi-step loss $\mathcal{L}_{\text{multi-step}}$ supervises all intermediate predictions, ensuring stable and accurate simulations.

emerges only after several steps. Multi-step objectives have been proposed to overcome the short-comings of this one-step training (Lienen & Günnemann, 2022). In practice, however, they only work over short horizons, since message passing quickly becomes unstable and loses information after a few steps (Arroyo et al., 2025), making direct long-horizon training infeasible. This limitation is fundamental to existing GNSs, as their GNN architectures struggle with long-range dependencies due to over-smoothing (Cai & Wang, 2020; Rusch et al., 2023), over-squashing (Alon & Yahav, 2021; Di Giovanni et al., 2023), and more generally gradient degradation (Arroyo et al., 2025).

We propose Information-preserving Graph Neural Simulators (IGNS), a neural simulator that learns continuous dynamics in a latent space and enables effective information propagation across the graph for accurate predictions. A schematic of IGNS is shown in Fig. 1; unlike existing approaches with dissipative updates that rapidly degrade information as message-passing depth increases, IGNS structures information flow through port-Hamiltonian dynamics (Van der Schaft, 2017; Galimberti et al., 2021; Lanthaler et al., 2023; Heilig et al., 2025), better preserving long-range dependencies between distant nodes. By training with a multi-step objective, our setup explicitly learns the underlying PDE by matching the solution of the parameterized port-Hamiltonian dynamics to ground-truth trajectories over a time window $T$, enforcing states to remain on-trajectory and thereby reducing error accumulation (Radler et al., 2025). Each integration step in IGNS thus mirrors a forward update of a classical PDE solver, where signals naturally propagate in space as time evolves.

In addition, IGNS incorporates a warmup phase to initiate global context before rollout and a geometric encoding module to capture local spatial relations on irregular meshes. Together, these components improve stability and generalization, enabling accurate simulation across diverse physical systems while maintaining a physically consistent inductive bias. On benchmarks targeting long-range and complex dynamics, IGNS achieves lower error and outperforms strong GNS baselines by staying on-trajectory over long horizons with high accuracy up to $T=200$. By contrast, prior work either (i) uses rather short supervision windows ($T \leq 20$) at train and test (Lienen & Günnemann, 2022; Shi et al., 2023b) or trains on short-windows and then rolls out autoregressively (Janny et al., 2023), or (ii) applies multi-step losses only in a reduced latent space rather than on the original graph (Han et al., 2022).

To summarize, we **(i)** introduce Information-preserving Graph Neural Simulators (IGNS), a graph neural simulator built on the port-Hamiltonian formalism that supports information-preserving roll-outs with high accuracy; **(ii)** analyze its theoretical properties, showing that IGNS maintains information flow across the spatial domain while capturing a broad class of dynamics; **(iii)** aim to mimic classical PDE solvers by matching, through a multi-step objective, the trajectory obtained by integrating the port-Hamiltonian core of IGNS with the ground-truth trajectory; and **(iv)** propose new tasks including Plate Deformation, Sphere Cloth, and Wave Balls, specifically designed to evaluate long-range propagation and oscillatory dynamics under external forcing.

## 2 BACKGROUND

In this section, we provide an overview of PDEs for modeling physical systems, followed by a brief discussion of message-passing-based GNSs for learning such systems. An in-depth discussion of

related works on GNSs, effective propagation, and Hamiltonian-inspired neural networks is provided in Appendix B.

## 2.1 PARTIAL DIFFERENTIAL EQUATIONS FOR PHYSICAL MODELING

Many physical phenomena can be described by PDEs, which relate the rates of change of a quantity with respect to both space and time. A general form of a time-dependent PDE is given by

$$\partial_t u = F(u, \nabla u, \nabla^2 u, \ldots; \boldsymbol{x}, t), \qquad (\boldsymbol{x}, t) \in \Omega \times (0, T], \tag{1}$$

where $u : \Omega \times [0, T] \to \mathbb{R}$ is the unknown function (without loss of generality, assumed to be a scalar field), $\Omega \subseteq \mathbb{R}^d$ is the spatial domain, and $F$ is a (possibly nonlinear) function of $u$ and its spatial derivatives.

To solve Eq. (1) numerically, one typically discretizes the spatial domain using methods such as finite differences or finite elements, resulting in a system of ordinary differential equations (ODEs) that can be integrated over time using various time-stepping schemes (Igel, 2016). This is often referred to as the method of lines, where spatial derivatives are approximated at discrete points, leading to a system of ODEs of the form

$$\dot{\boldsymbol{u}}_i(t) = \boldsymbol{f}_i(t, \boldsymbol{u}(t)), \quad i = 1, \ldots, n, \quad \boldsymbol{u}(0) = \bar{\boldsymbol{u}}, \tag{2}$$

where $\boldsymbol{u}(t) = [\boldsymbol{u}_1(t), \ldots, \boldsymbol{u}_n(t)]^\top$ represents the state of the system at discrete spatial points, and $\boldsymbol{f}_i$ encapsulates the spatial discretization of $F$ at point $i$. The initial condition $\bar{\boldsymbol{u}}$ is derived from the initial state of the PDE.

**Remark.** While first-order systems as in Eq. (1) are common, many physical phenomena are better captured by second-order dynamical systems (Evans, 2010), which naturally arise in the modeling of oscillatory behavior like propagation of waves and vibrations. A general second-order PDE with damping and external forcing can be written as

$$\partial_{tt} u + \mathcal{B} \, \partial_t u + \mathcal{L} \, u = g(\boldsymbol{x}, t), \qquad (\boldsymbol{x}, t) \in \Omega \times (0, T],$$

where $\mathcal{L}$ and $\mathcal{B}$ are linear operators on $L^2(\Omega)$, and $g$ is a prescribed forcing term. A detailed derivation of applying the method of lines to this equation is provided in Appendix C.2, leading to a system of second-order ODEs.

## 2.2 MESSAGE PASSING FOR MODEL LEARNING

Under the model learning settings, the goal is to learn the underlying dynamics (i.e., the RHS of Eq. (1)) from data. Similar to traditional numerical methods, we first need to discretize the spatial domain into a set of points or elements. This discretization can be naturally represented as a graph, $G = (V, E)$ where $V$ is the set of $n$ nodes (i.e., the mesh entities) and $E \subseteq V \times V$ is the set of $m$ edges capturing their interactions. Then, following the method of lines, we obtain the following system of ODEs:

$$\dot{\boldsymbol{x}}_i^{(t)} = \boldsymbol{f}_i(t, \boldsymbol{X}^{(t)}; \boldsymbol{E}; G), \quad i = 1, \ldots, n, \quad \boldsymbol{X}^{(0)} = \bar{\boldsymbol{X}}, \tag{3}$$

where $\boldsymbol{x}_i^{(t)} \in \mathbb{R}^d$ is the state vector of node $i$ at time $t$, $\boldsymbol{X}(t) = [\boldsymbol{x}_1^{(t)}, \ldots, \boldsymbol{x}_n^{(t)}]^\top \in \mathbb{R}^{n \times d}$ is the node-state matrix, $\boldsymbol{E} = [\boldsymbol{e}_{ij}, \ldots]^T \in \mathbb{R}^{m \times c}$ is the edge-feature matrix, and $\boldsymbol{f}_i$ is the unknown function that captures the interactions between nodes based on the graph structure $G$. Here, we also assume the initial condition $\bar{\boldsymbol{X}}$ is given.

Solving this requires integrating Eq. (3) over time, which can be done using standard ODE solvers (Hairer et al., 1993). However, standard Graph Neural Simulators (GNSs) simplify this process by further discretizing time using a fixed step size $\Delta t$ and applying the message-passing paradigm to compute the node states at the next time step $\boldsymbol{x}^{(t+1)}$, leading to the following update rule

$$\boldsymbol{x}_i^{(t+1)} = \boldsymbol{x}_i^{(t)} + \Delta t \, \boldsymbol{g}_\theta^L(\boldsymbol{X}^{(t)}; \boldsymbol{E}; G). \tag{4}$$

Here, $\boldsymbol{g}_\theta^L$ denotes the function obtained by stacking $L$ message-passing layers, which aggregate information from neighboring nodes to update the state of each node, defined as

$$\boldsymbol{x}_i' = \phi_{\text{node;}}\left(\boldsymbol{x}_i, \sum_{j \in \mathcal{N}_i} \boldsymbol{e}_{ij}'\right), \quad \boldsymbol{e}_{ij}' = \phi_{\text{edge}}(\boldsymbol{x}_i, \boldsymbol{x}_j, \boldsymbol{e}_{ij}). \tag{5}$$

Here, $\phi_{\text{edge}}$ is the message function, $\phi_{\text{node}}$ is the node update function, and $\mathcal{N}_i$ is the set of neighbors of node $i$. Both $\phi_{\text{edge}}$ and $\phi_{\text{node}}$ are often implemented as MLPs. We can see that under this setting, $\boldsymbol{f}_i$ in Eq. (3) is approximated by a stack of $L$ message-passing layers, i.e., $\boldsymbol{f}_i(t, \boldsymbol{X}^{(t)}; \boldsymbol{E}; G) \approx \boldsymbol{g}_\theta^L(\boldsymbol{X}^{(t)}; \boldsymbol{E}; G)$. In practice, one can obtain the whole trajectory by iteratively applying Eq. (4) for $T$ time steps, starting from the initial condition $\bar{\boldsymbol{X}}$. However, this approach often suffers from error accumulation over long time horizons, limiting long-term accuracy (Brandstetter et al., 2022).

In this work, guided by the theory of dynamical systems, we propose a novel GNS that directly models the underlying continuous dynamics in Eq. (3), built upon the port-Hamiltonian formalism. Under this framework, information is maintained more effectively across the entire space. Combined with the multi-step loss, the learned model demonstrates that it can mitigate the error accumulation issue and achieve greater accuracy on long rollouts. We discuss them in detail in the next section.

## 3 THE INFORMATION-PRESERVING GRAPH NEURAL SIMULATORS

**Problem Statement.** Given a time series of node states $\{\boldsymbol{X}^{(t)}\}_{t=0}^T$ on an irregular mesh or graph $G = (V, E)$, our goal is to learn and simulate the evolution of each node $\boldsymbol{x}_i^{(t)}$ over time. To address this, we propose the Information-preserving Graph Neural Simulators (IGNS), a GNN-based simulator that uses port-Hamiltonian dynamics as its latent evolution core. Fig. 1 shows the high-level overview of our method. IGNS is built on four key components:

- *Port-Hamiltonian formalism*: by parameterizing a Hamiltonian $H$ and adding damping and external forcing terms, IGNS captures both energy-conserving and non-conservative behavior.
- *Warmup*: the conservation property is further exploited through the use of a *warmup phase* at initialization, enabling the model to obtain a more global context from the start of the rollout.
- *Geometric encoding*: by embedding geometric features directly into the edge attributes, IGNS can operate on irregular meshes and exploit spatial structure for more accurate local interactions.
- *Multi-step objective*: it supervises the entire rollout windows, reducing error accumulation, improving data efficiency, and better aligning training with long-horizon prediction.

### 3.1 PORT-HAMILTONIAN FORMALISM

We start by expressing the dynamics on the RHS of the Eq. (3) using the port-Hamiltonian formalism (van der Schaft & Jeltsema, 2014; Desai et al., 2021; Heilig et al., 2025), which is a generalization of Hamiltonian systems that integrate both conservative and non-conservative dynamics based on energy principles. Specifically, we parameterize the Hamiltonian $H_\theta(t, \boldsymbol{X})$ and evolve the joint state according to

$$\dot{\boldsymbol{x}}_i = \boldsymbol{J}\nabla_{\boldsymbol{x}_i}H_\theta(t, \boldsymbol{X}) - \underbrace{\begin{bmatrix} 0 \\ \boldsymbol{D}_\theta\nabla_{\boldsymbol{p}_i}H_\theta(t, \boldsymbol{X}) \end{bmatrix}}_{\text{damping}} + \underbrace{\begin{bmatrix} 0 \\ \boldsymbol{r}_\theta(t, \boldsymbol{X}) \end{bmatrix}}_{\text{forcing \& residual}}, \quad \boldsymbol{J} = \begin{bmatrix} 0 & \boldsymbol{I} \\ -\boldsymbol{I} & 0 \end{bmatrix}. \quad (6)$$

Here, we omit the time index $t$ for clarity, and denote $\boldsymbol{x}_i = [\boldsymbol{q}; \boldsymbol{p}]_i^T$ with $\boldsymbol{q} \in \mathbb{R}^{n \times \frac{d}{2}}$ and $\boldsymbol{p} \in \mathbb{R}^{n \times \frac{d}{2}}$ to represent the generalized coordinates and momenta, respectively. Without forcing and damping, Eq. (6) reduces to the classical Hamiltonian dynamics, which conserves the energy $H_\theta$ over time. However, most real-world physical systems are non-conservative, where energy can be dissipated (e.g., due to friction) or injected (e.g., due to external forces). To account for this, two additional terms are introduced: $\boldsymbol{D}_\theta\nabla_{\boldsymbol{p}_i}H_\theta(t, \boldsymbol{X})$ and $\boldsymbol{r}_\theta(t, \boldsymbol{X})$, for damping and external forcing, respectively. Next, we parameterize the Hamiltonian $H_\theta$ as

$$H_\theta(t, \boldsymbol{X}) = \sum_{i \in \mathcal{V}} \tilde{\sigma}\left(\boldsymbol{W}(t)\boldsymbol{x}_i + \sum_{j \in \mathcal{N}_i} \boldsymbol{V}(t)\boldsymbol{x}_j\right)^T \cdot \boldsymbol{1}_d, \quad (7)$$

where $\tilde{\sigma}$ is an anti-derivative of a non-linear activation function $\sigma$, $\boldsymbol{1}_d$ is a row vector of ones of dimension $d$, and $\boldsymbol{W}(t)$ and $\boldsymbol{V}(t)$ are block diagonal matrices (to ensure the separation into the $\boldsymbol{q}$ and $\boldsymbol{p}$ components), i.e., $\boldsymbol{W}(t) = \text{diag}([\boldsymbol{W}_{\boldsymbol{q}}(t), \boldsymbol{W}_{\boldsymbol{p}}(t)])$ and $\boldsymbol{V}(t) = \text{diag}([\boldsymbol{V}_{\boldsymbol{q}}(t), \boldsymbol{V}_{\boldsymbol{p}}(t)])$. The matrices $\boldsymbol{W}(t)$ and $\boldsymbol{V}(t)$ can be either shared across time (i.e., $\boldsymbol{W}(t) = \boldsymbol{W}$ and $\boldsymbol{V}(t) = \boldsymbol{V} \; \forall t$) or time dependent, as further discussed in Appendix D.

To justify this choice of dynamics, we provide a more theoretical discussion in Section 4, showing that Eq. (6) is universal, and via energy conservation, it allows stable long-range information propagation across space. Hence, it is well-suited for modeling complex physical systems.

**Symplectic Integrator.** To ensure the energy-conserving property of the Hamiltonian dynamics, we employ a symplectic integrator to numerically solve Eq. (6). Specifically, we choose the symplectic Euler method (Hairer et al., 2006), which updates the state as follows:

$$
\begin{aligned}
\boldsymbol{p}_i^{(t+1)} &= \boldsymbol{p}_i^{(t)} + \Delta t\big( -\nabla_{\boldsymbol{q}_i} H_\theta(t, \boldsymbol{q}^{(t)}, \boldsymbol{p}^{(t)}) - \boldsymbol{D}_\theta \nabla_{\boldsymbol{p}_i} H_\theta(t, \boldsymbol{q}^{(t)}, \boldsymbol{p}^{(t)}) + \boldsymbol{r}_\theta(t, \boldsymbol{X}^{(t)})\big), \\
\boldsymbol{q}_i^{(t+1)} &= \boldsymbol{q}_i^{(t)} + \Delta t \nabla_{\boldsymbol{p}_i} H_\theta(t, \boldsymbol{q}^{(t)}, \boldsymbol{p}^{(t+1)}).
\end{aligned}
\tag{8}
$$

Here, the gradients of the Hamiltonian are computed with respect to the generalized coordinates and momenta.

## 3.2 WARMUP PHASE

Under the message-passing framework, a single update propagates information only to direct neighbors. This locality is a fundamental limitation when simulating physical systems, where long-range interactions often play a crucial role from the very first timestep. For example, in mesh-based simulations of fluids or elastic materials, the evolution of a node may depend not only on its immediate surroundings but also on distant regions of the mesh, such as boundary conditions or sources of external forces. If the model starts from a completely local representation, it must accumulate long-range information gradually over several rollout steps, which delays the emergence of globally consistent dynamics and may degrade accuracy in the early stages of the simulation.

To alleviate this issue, we introduce a *warmup phase* applied once at initialization. Starting from the initial state, we perform $l$ additional rounds of message passing without advancing time, as visually summarized on the left of Fig. 1. More formally, the initial condition of the simulation is set to $\boldsymbol{X}^{(0)} = \bar{\boldsymbol{X}}^{(l)}$, where $\bar{\boldsymbol{X}}^{(l)}$ denotes the state after $l$ warmup iterations. This strategy broadcasts information across the graph up to a radius $l$ from each node, so that for sufficiently large $l$, each node effectively receives signals from increasingly larger neighborhoods. Thanks to the energy-conserving core of IGNS, this globally informed latent state is preserved throughout the rollout, rather than being dissipated. The effect of $l$ is further analyzed in Section 6.

## 3.3 GEOMETRICAL INFORMATION ENCODING

Accurately encoding geometry is essential for modeling physical systems on irregular meshes, where interactions depend on relative positions (i.e., in the mesh space) rather than absolute states (i.e., in the physical world space). To capture this relationship, we model the external force to drive the system's evolution based on geometrical information, thereby allowing the dynamics to be influenced by the spatial structure. We design the external force following an edge-level message-passing scheme inspired by Pfaff et al. (2021), where each edge feature $\boldsymbol{e}_{ij}$ encodes both the displacement vectors ($\boldsymbol{s}_{ij} = \text{pos}_j - \text{pos}_i$) and distances ($\boldsymbol{d}_{ij} = \|\text{pos}_j - \text{pos}_i\|_2$), with $\text{pos}_i$ indicating the position of node $i$ in the mesh space. More formally, we formulate the external force as $\boldsymbol{r}_\theta(t, \boldsymbol{X}) = \boldsymbol{g}_\theta^L(\cdot)$ where each message-passing step is defined as

$$
\phi_{\text{node}}(\cdot) = \sigma\left(\boldsymbol{W}_{\text{node}}\boldsymbol{q}_i + \sum_{j \in \mathcal{N}_i} \boldsymbol{e}'_{ij}\right), \quad \boldsymbol{e}'_{ij} = \phi_{\text{edge}}(\cdot) = \boldsymbol{W}_{\text{edge}}(\boldsymbol{q}_j - \boldsymbol{q}_i) + \boldsymbol{e}_{ij}
\tag{9}
$$

with $\boldsymbol{e}_{ij}$ as the concatenation between displacement vectors and distance values, i.e., $\boldsymbol{e}_{ij} = [\boldsymbol{s}_{ij}, \boldsymbol{d}_{ij}]$. This formulation explicitly separates the edge in the world space ($\boldsymbol{q}_j - \boldsymbol{q}_i$) and the edge in the mesh space ($\boldsymbol{e}_{ij}$) in the edge update. Unlike Pfaff et al. (2021), we do not update edge messages at every time step. Combined with the multi-step loss (discussed below), the edge information serves instead as a static prior that weighs neighbor messages. This makes the model less dependent on explicit geometry, helps prevent overfitting to specific meshes, and still provides sufficiently useful context. We empirically study this hypothesis in Appendix I.

## 3.4 MULTI-STEP LOSS

We train our models through a multi-step objective. Given a window $(\boldsymbol{q}^{(t)}, \ldots, \boldsymbol{q}^{(t+K)})$, we roll out from $\boldsymbol{q}^{(t)}$ and optimize both the predicted $\widehat{\boldsymbol{q}}^{(t+\tau)}$ and $\widehat{\boldsymbol{p}}^{(t+\tau)}$ to match its corresponding ground truth

fields $\boldsymbol{q}^{(t+\tau)}$ and velocities $\boldsymbol{p}^{(t+\tau)}, \forall \tau \in [1, K]$,

$$\mathcal{L}_{\text{multi-step}} = \sum_{\tau=1}^{K} \left( \|\widehat{\boldsymbol{q}}^{(t+\tau)} - \boldsymbol{q}^{(t+\tau)}\|_2^2 + \|\widehat{\boldsymbol{p}}^{(t+\tau)} - \boldsymbol{p}^{(t+\tau)}\|_2^2 \right). \tag{10}$$

This multi-step loss enhances the consistency between the prediction and ground-truth data on a trajectory level. In contrast, one-step training followed by autoregressive rollout is prone to suffering from the error accumulation problem, especially when the time-series data exhibits a strong temporal correlation, which we analyze in Section 6. The multi-step loss also fits IGNS well because the non-dissipative core preserves signals across time; therefore, the gradients from distant steps do not vanish, and the model can make use of the whole trajectory. Standard first-order message passing instead tends to oversmooth, gradients decay over many steps, and the benefit of a long horizon objective is limited. Finally, including $\boldsymbol{p}$ helps improve performance further; when momenta are not explicit in the state, we approximate them by finite differences on the ground truth $\boldsymbol{q}^{(t)}$ and $\boldsymbol{q}^{(t+\tau)}$. A visual exemplification of this multi-step loss strategy is summarized on the right of Fig. 1.

## 4 THEORETICAL PROPERTIES

In this section, we discuss the theoretical benefits of our IGNS, focusing on universality and effective information propagation during the simulation, underpinning its ability to effectively simulate complex physical systems. A more in-depth discussion is provided in Appendix E.

**Universality.** We start by showing that our IGNS, with state updates in Eq. (6) and Hamiltonian in Eq. (7) yields a universal model. Therefore, IGNS can approximate any given functional $\Phi : \mathbb{R}^{n \times d} \to \mathbb{R}^{n \times d}$ that maps the initial condition to the solution at time $\tau$.

**Theorem 1.** *Let $\Psi_\theta$ be the functional that maps the initial condition $\boldsymbol{x}_0 = [\boldsymbol{q}_0, \boldsymbol{p}_0]$ to the solution at time $\tau$ to the ODE defined Eq. (6) and Eq. (7). In other words, $\Psi_\theta$ represents IGNS. Then, for any $F : \mathbb{R}^{n \times d} \to \mathbb{R}^{n \times d}$ with compact support, which represents the evolution of the system after time $\tau$, there exist a set of parameters $\theta$ such that $\|\Psi_\theta(\boldsymbol{x}_0) - F(\boldsymbol{x}_0)\| \leq \epsilon$.*

The proof is reported in Appendix E.1, along with an in-depth discussion and interpretation for physical simulations. As a consequence of the theorem and its proof, IGNS is capable of approximating any functional with compact support that maps the time-dependent signal of external forces and geometrical information into the dynamics of the physical system, making it suitable for accurately modeling complex interactions and dynamics in arbitrary graph-structured systems.

**Remark 1.** *Compactness in the support of the functional is a common requirement for Universality theorems. However, this is not restrictive for physical systems, as it is sufficient to have a limited domain and upper bounds on the physical quantities and energy in the system to satisfy this assumption. All these conditions are perfectly reasonable in contained physical simulations. See Appendix E.1 for a more in-depth discussion on this topic.*

**Long-Range Information Propagation.** Our next focus is on the information propagation capability of IGNS, a critical aspect for physical simulations, since reliable modeling hinges on the ability to capture long-range interactions in both time and space.

In the absence of damping and external forces, our IGNS reduces to a pure Hamiltonian systems and thus adheres to the corresponding conservation laws (Galimberti et al., 2021; 2023; Heilig et al., 2025). Consequently, the vector field induced by the conservative IGNS is divergence-free, ensuring information preservation throughout propagation. In other words, under this setting, the system dynamics can be interpreted as purely rotational, with no information dissipation.

Following Arroyo et al. (2025), which demonstrates that the inability to perform effective propagation is closely tied to the vanishing gradient problem, we investigate the behavior of IGNS with the Hamiltonian proposed in Eq. (7) through a sensitivity analysis. Thus, we study the quantity $\|\partial \boldsymbol{x}(t)/\partial \boldsymbol{x}(s)\|$, as formalized in the following theorem, which follows the sensitivity arguments developed in Heilig et al. (2025).

**Theorem 2.** *If $\boldsymbol{x}(t)$ is the solution to the ODE $\dot{\boldsymbol{x}}_i = \boldsymbol{J} \nabla_{\boldsymbol{x}_i} H_\theta(t, \boldsymbol{X})$, that is, the Hamiltonian dynamic defined in Eq. (6) without dampening and external forcing, then $\left\| \frac{\partial \boldsymbol{x}(t)}{\partial \boldsymbol{x}(s)} \right\| \geq 1$ for any $0 \leq s \leq t$.*

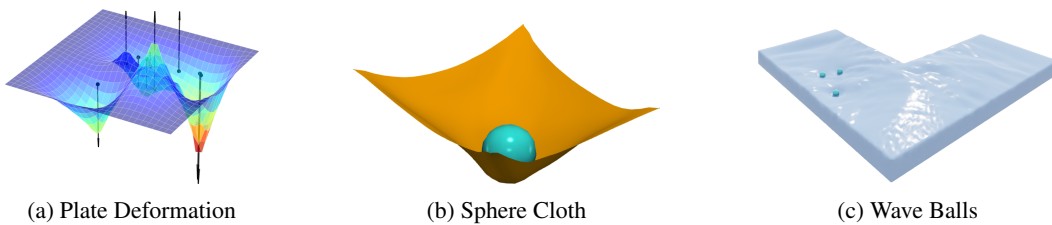

(a) Plate Deformation        (b) Sphere Cloth        (c) Wave Balls

Figure 2: Three novel long-range simulation tasks. *(a)* **Plate Deformation**: a flat plate deformed subject to varying numbers and magnitudes of point forces. *(b)* **Sphere Cloth**: a cloth mesh impacted by a falling sphere, producing elastic deformation with contact dynamics. *(c)* **Wave Balls**: surface waves on water generated by three balls moving linearly from different initial positions.

The proof is discussed in Appendix E.2. This theorem shows that the sensitivity matrix is lower bounded by a constant, ensuring non-vanishing gradients and thereby enabling effective long-range information propagation; unlike GCNs (Kipf & Welling, 2017), which suffer from an exponential decay in propagation rate over time (Gravina et al., 2025). This non-dissipative behavior also enables IGNS to propagate and preserve a broader context across nodes during the warmup phase (see Section 3.2).

Although Hamiltonian dynamics provide theoretical guarantees for propagation behavior, additional terms, such as dissipation and external forcing, may counteract this effect. To this end, we recall the discussion in Rusch et al. (2022); there, they showed that the oscillatory behavior of the system allows the gradients not to decay exponentially even in the presence of these additional terms. More specifically, in the limit of small $\Delta t$, the gradient of the loss function with respect to any learnable weight parameter in the forcing term is independent of the number of updates. We formalize this with a statement in Appendix E.3, inspired from Proposition 3.6 of Rusch et al. (2022). Additionally, as long as the dependence between $\Delta t$ and the number of iterations $N$ is polynomial $\Delta t \sim N^{-s}$, gradients do not decay exponentially in $N$, alleviating the vanishing gradient problem.

## 5 EXPERIMENTAL SETUP

**Datasets.** We evaluate our methods on six benchmark datasets in two categories: Lagrangian systems (*Plate Deformation*, *Impact Plate*, *Sphere Cloth*) and Eulerian systems (*Wave Balls*, *Cylinder Flow*, *Kuramoto-Sivashinsky*). First, we introduce **Plate Deformation** (Plate Def.) task to evaluate long-range propagation effects on spatial domain, ignoring the evolution over time. Here, given only force positions and magnitudes, the model must learn to predict the final deformed state of an initially flat sheet. Next, the *Impact Plate* task, proposed in HCMT (Yu et al., 2024), tests the model's ability to capture rapid stress propagation across a large mesh over time. We then introduce two additional new tasks, **Sphere Cloth** and **Wave Balls**, both designed to evaluate oscillatory behavior under external forcing. Finally, we include two complex dynamics tasks, *Cylinder Flow* (Pfaff et al., 2021) and the fourth-order *Kuramoto-Sivashinsky* (KS) equation, each slightly modified to probe error accumulation under autoregressive training. Fig. 2 highlights the *three novel tasks* we introduced. Full dataset specifications and simulator details are in Appendix F (Table 4).

**Evaluating Methods.** We evaluate a broad set of methods, grouped into standard graph-convolution (Graph Conv.), effective propagation (Eff. Propagation), and our proposed port-Hamiltonian (Port-Ham.) models. **Graph Conv.:** Our main competitor, *MGN* (MeshGraphNets) (Pfaff et al., 2021), is a standard message passing GNN trained autoregressively with explicit Euler integration. For the remaining baselines, we use the multi-step loss (Eq. (10)). *MP-ODE* (Iakovlev et al., 2021; Lienen & Günnemann, 2022) applies neural ODE integration each step; here, we choose the RK4 integrator (instead of `dopri5`) to ensure a fair training budget compared to other baselines; additionally, we define the message function on nodal points instead of cell elements as in Lienen & Günnemann (2022). Next, *GCN+LN* is a graph convolutional network with LayerNorm, a common technique to mitigate oversmoothing in deep message passing (Luo et al., 2024). **Eff. Propagation**: *ADGN* (Gravina et al., 2023) adopted an anti-symmetric matrix in the graph convolution to avoid dissipative behavior, *GraphCON* (Rusch et al., 2022) is an oscillatory graph network, originally proposed to avoid oversmoothing effects. **Port-Ham. (Ours)**: *IGNS$_{ti}$* (time independent) and *IGNS* (time varying) are port-Hamiltonian-based models that follow the state update in Eq. (6). For

Table 1: Per–task test MSE (mean $\pm$ std, lower is better). Methods are grouped by operator class, *(I)* Graph Conv., *(II)* Eff. Propagation and *(III)* **Port-Ham. (Ours)**. Results are averaged over 4 seeds. Best and second best results are highlighted in **orange** and **teal**, respectively.

|  | Method | Plate Def. MSE | Impact Plate MSE | Sphere Cloth MSE ($\times 10^{-3}$) | Wave Balls MSE ($\times 10^{-3}$) | Cylinder Flow MSE ($\times 10^{-3}$) | KS MSE ($\times 10^{-3}$) |
|---|---|---|---|---|---|---|---|
| *(I)* | GCN+LN | $3.98 \pm 0.66$ | $3997.98 \pm 199.83$ | $26.45 \pm 0.23$ | $1.95 \pm 0.14$ | $11.71 \pm 1.03$ | $3.10 \pm 0.78$ |
|  | MGN | $1.27 \pm 0.06$ | $3095.75 \pm 908.58$ | $32.07 \pm 2.45$ | $1.78 \pm 0.14$ | $12.08 \pm 2.60$ | $10.76 \pm 9.16$ |
|  | MP-ODE | $-$ | $-$ | $25.75 \pm 1.32$ | $97.04 \pm 0.70$ | $18.17 \pm 1.15$ | $59.55 \pm 14.82$ |
| *(II)* | A-DGN | $1.99 \pm 0.76$ | $3974.56 \pm 309.98$ | $30.99 \pm 0.80$ | $0.70 \pm 0.05$ | $14.72 \pm 0.94$ | $0.97 \pm 0.89$ |
|  | GraphCON | $1.20 \pm 0.02$ | $2279.09 \pm 720.25$ | $29.00 \pm 0.44$ | $1.62 \pm 0.03$ | $7.32 \pm 0.05$ | $0.49 \pm 0.02$ |
| *(III)* | **IGNS$_{\text{ti}}$** | $1.43 \pm 0.04$ | $75.99 \pm 17.17$ | $28.20 \pm 0.35$ | $0.45 \pm 0.01$ | $9.12 \pm 0.40$ | $0.86 \pm 0.12$ |
|  | **IGNS** | $1.34 \pm 0.05$ | $67.86 \pm 14.75$ | $24.16 \pm 0.67$ | $0.37 \pm 0.01$ | $8.21 \pm 0.38$ | $1.20 \pm 0.18$ |

fairness, all baselines, except *MGN* which updates edge features at each message-passing step, use the same proposed geometric encoding (Section 3.3). Appendix G provides details on the choice of hyperparameters and training budget.

Our experimental setup highlights the importance of being on-trajectory during rollout via the use of multi-step loss, i.e., to avoid the over-reliance on autoregressive state carry-over (Brandstetter et al., 2022; Radler et al., 2025). Therefore, a single window $T$ is fixed for both training and evaluation.

## 6 RESULTS AND DISCUSSIONS

Table 1 summarizes the main results across all six tasks, reporting mean and standard deviation of test MSE over 4 random seeds. Overall, IGNS$_{\text{ti}}$ and IGNS consistently outperform the main competitor MGN, as well as other baselines, across all tasks. Notably, GraphCON with the geometric encoding also performs strongly on static graph tasks (Plate Def., Cylinder Flow, KS) yet fails severely on the remaining ones. This suggests that while the oscillatory bias in GraphCON improves over standard graph convolution methods, its formulation limits its expressiveness for more complex dynamics. To gain further insight, we provide an analysis of the relationship between GraphCON and our models in Appendix C. Furthermore, introducing time-variation in the weight matrices (IGNS) produces further gains, suggesting that flexible, time-dependent parameterizations better capture complex, nonstationary dynamics. We now analyze per-task behavior below.

Table 2: Comparisons with additional strong baselines on Plate Deformation, Impact Plate and Kuramoto-Sivashinsky tasks. Best results are highlighted in **orange**.

| Method | MSE |
|---|---|
| MGN-rewiring | $3.72 \pm 0.15$ |
| MGN-shared | $8.21 \pm 1.98$ |
| MGN | $1.27 \pm 0.06$ |
| **IGNS$_{\text{ti}}$** | $1.43 \pm 0.04$ |
| **IGNS** | $1.34 \pm 0.05$ |

(a) Plate Deformation

| Method | MSE |
|---|---|
| HCMT | $646.81 \pm 27.87$ |
| MGN | $3095.75 \pm 908.58$ |
| **IGNS$_{\text{ti}}$** | $75.99 \pm 17.17$ |
| **IGNS** | $67.86 \pm 14.75$ |

(b) Impact Plate

| Method | MSE ($\times 10^{-3}$) |
|---|---|
| FNO-RNN | $6.69 \pm 0.46$ |
| MGN | $10.76 \pm 9.16$ |
| **IGNS$_{\text{ti}}$** | $0.86 \pm 0.12$ |
| **IGNS** | $1.20 \pm 0.18$ |

(c) Kuramoto-Sivashinsky

**Long-range propagation.** In Plate Deformation (final-state prediction only), the port-Hamiltonian-based models (IGNS$_{\text{ti}}$, IGNS) and GraphCON achieve low error, while A-DGN and GCN+LN perform substantially worse. Interestingly, MGN also performs well on this task. To better understand this result, we further evaluate two MGN variants: MGN-shared, which uses a shared processor across steps, and MGN-rewiring, which connects each force node to every second node in the mesh. As shown in Table 2 (a), both variants yield higher error than standard MGN, suggesting that MGN's strong performance is primarily due to its non-shared processor. Although this design increases the parameter count significantly ($\times$#MP_Layers), it allows the model to overfit to geometric details. Additional visualizations supporting this observation are provided in Appendix I.

For Impact Plate, standard MGN fails by a large margin, whereas IGNS attains the lowest error, outperforming HCMT by nearly an order of magnitude. This result highlights that the energy-

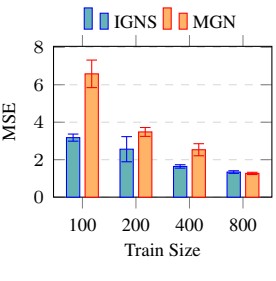 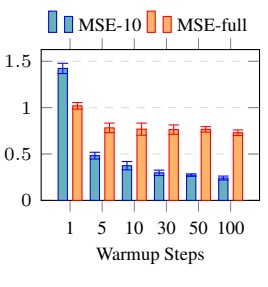 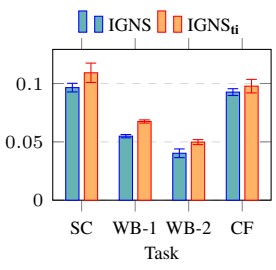

(a) Data efficiency.    (b) Varying warmup steps.    (c) Longer horizon.

Figure 3: Ablation studies. *(a)* Data efficiency on the **final-state-only** Plate Deformation task. *(b)* Effect of warmup steps on the **full-rollout** Plate Deformation task. *(c)* Longer-horizon performance on Sphere Cloth (SC) with $T = 100$, Wave Balls (WB) with two variants: WB-1 ($T = 100$) and WB-2 ($T = 200$), and Cylinder Flow (CF) with $T = 100$. For better visualization in the bar plots, MSE values for Wave Balls and Cylinder Flow are scaled by $\times 100$ and $\times 10$, respectively. All results are averaged over 4 seeds.

preserving property of Hamiltonian dynamics enables effective long-range transmission of elastic energy between distant nodes, without requiring specialized hierarchical architectures. Notably, we find that GraphCON's loss consistently diverges after a certain number of training iterations across all seeds, indicating that its fixed oscillatory bias may be too restrictive to capture the dynamics present in this task. Note that we omit MP-ODE results on these two tasks, as the method was not proposed for static graphs, and for Impact Plate, it fails to converge during training.

**Complex dynamics analysis.** We now analyze performance across the remaining tasks in Table 1, that involve complex dynamics and oscillatory behavior. For Sphere Cloth, IGNS achieves the lowest error, while the other models yield errors of similar magnitude, with MGN exhibiting the largest error. In Wave Balls, our proposed models (IGNS, IGNS$_{\mathbf{ti}}$) significantly outperform all baselines, suggesting that the port-Hamiltonian structure, viewed as a generalization of wave equations (Appendix C), is particularly well-suited for this type of wave dynamics.

For Cylinder Flow, both port-Hamiltonian-based and GraphCON models perform comparably, while A-DGN, GCN+LN, and MP-ODE show larger errors under the causal-only setup used here. Notably, in the original paper (Pfaff et al., 2021), MGN performs better due to longer autonomous portions in the dataset; our causal-focused split reduces that advantage (Appendix F). Finally, for Kuramoto-Sivashinsky, the chaotic dynamics heavily penalize autoregressive MGN (resulting in large error), while GraphCON, IGNS$_{\mathbf{ti}}$/IGNS, and A-DGN remain comparatively strong; GCN+LN struggles to represent the emergence of high-frequency modes towards the end of the evolution. Although Fourier Neural Operators (FNO-RNN) Li et al. (2021) often perform well on this type of dynamic, they underperform significantly compared to graph-based models, as shown in Table 2 (c). This is likely because, unlike the usual setup with uniformly random initial states, here we initialize with three Gaussian sources randomly placed in the grid to encourage local propagation. Further discussion and visualizations for all tasks are provided in Appendix I.

Table 3: Ablations of IGNS on *SphereCloth-direct*. We report test MSE, values scaled by $\times 10^{-3}$. (a) Core components: geometric encoding, warmup, and multi-step loss. (b) Damping and forcing in IGNS. (c) Damping and forcing in IGNS$_{\mathbf{ti}}$.

| Method | MSE ($\times 10^{-3}$) |
|---|---|
| IGNS$_{\text{no-geo}}$ | $36.4 \pm 3.0$ |
| IGNS$_{\text{no-warmup}}$ | $27.6 \pm 0.5$ |
| IGNS$_{\text{no-multi-step}}$ | $2388.6 \pm 981.0$ |

(a) Core components

| Method | MSE ($\times 10^{-3}$) |
|---|---|
| IGNS$_{\text{no-}d,\text{no-}f}$ | $36.4 \pm 1.7$ |
| IGNS$_{\text{no-}d}$ | $25.6 \pm 0.3$ |
| IGNS$_{\text{no-}f}$ | $34.2 \pm 0.8$ |
| **IGNS** | $25.3 \pm 1.0$ |

(b) IGNS variants

| Method | MSE ($\times 10^{-3}$) |
|---|---|
| IGNS$_{\mathbf{ti},\text{no-}d,\text{no-}f}$ | $80.6 \pm 4.3$ |
| IGNS$_{\mathbf{ti},\text{no-}d}$ | $27.8 \pm 0.5$ |
| IGNS$_{\mathbf{ti},\text{no-}f}$ | $60.5 \pm 16.4$ |
| **IGNS$_{\mathbf{ti}}$** | $28.2 \pm 0.8$ |

(c) IGNS$_{\mathbf{ti}}$ variants

**Ablation on IGNS components.** We ablate IGNS's components on a harder variant, *SphereCloth-direct*, where the ball connects only to the four cloth corners (the original task adds extra connections

to every-four nodes of the cloth). This stresses long-range propagation: signals must travel from the ball through sparse corners before spreading across the cloth. Table 3 (a) shows that removing geometric encoding or the multi-step objective prevents reliable learning on this task, showing these two pieces are necessary for PDE matching; by contrast, dropping the warmup causes only a minor degradation, and we study it further below. We then ablate the port-Hamiltonian core on Table 3 (b) and (c): removing damping has only a minor effect, whereas removing external forcing severely degrades performance. This highlights the central role of forcing in driving information through the graph and supports our universality result in Appendix E.1. We extend this discussion with additional diagnostics and results in Appendix I.

**Additional Ablation Studies.** Runtime analysis is in Appendix H. **(a) Data efficiency.** We compare IGNS and MGN on Plate Deformation with varying training set sizes (100, 200, 400 and 800 samples). As shown in Fig. 3a, IGNS consistently outperforms MGN across all dataset sizes, with the performance gap widening as the training set decreases. This suggests that the inductive bias from port-Hamiltonian dynamics helps IGNS generalize better from limited data, while MGN leads to geometric overfitting when data is scarce. **(b) Warmup steps.** Next, we study the effect of varying the number of warmup steps $l$ used during training for IGNS on Plate Deformation with full-rollout prediction. Fig. 3b shows two metrics: cumulative MSE over the first 10 steps (MSE-10) and full-rollout validation loss. Both improve as $l$ increases. Moving from $l = 1$ to $l = 5$ yields the largest gain (since at $t = 1$, the task already requires global propagation across the plate); after $l = 30$ the full-rollout loss largely plateaus, while MSE-10 continues to improve with larger $l$, indicating larger warmup reduces early error directly, while smaller $l$ forces the model to learn dynamics that recover from early mistakes later (visualizations are shown in Fig. 13). **(c) Longer horizon.** Finally, we compare IGNS and IGNS$_\text{ti}$ on a longer rollout horizon setting for Sphere Cloth ($T = 100$), Wave Balls ($T = 100$, $T = 200$), and Cylinder Flow ($T = 100$) vs. $T = 50$ (Sphere Cloth, Wave Balls) and $T = 30$ (Cylinder Flow) in the main experiments. In Sphere Cloth, we additionally increase the number of nodes to $29 \times 29$ (vs. $19 \times 19$ in the main experiment) to increase complexity. For Wave Balls, due to the complexity of the task, we follow the curriculum scheduling proposed by Han et al. (2022); Lienen & Günnemann (2022). Here, we linearly increase the supervised window by $s$ steps from a minimal window $T_0$ to $T$ after every $n$ epochs. $T_0$, $s$ and $n$ are set as hyperparameters. Meanwhile, for Cylinder Flow with $T = 100$, we adopt the attentional aggregation (Li et al., 2019b) in the forcing term to further enhance expressiveness. As shown in Fig. 3c, IGNS outperforms IGNS$_\text{ti}$ on those tasks, with lower std. These results indicate that making weighted matrices time-varying helps increase the model's expressiveness and therefore, enables it to capture more complex dynamics.

## 7 CONCLUSION

In this work, we presented IGNS, a graph-based neural simulator that leverages port-Hamiltonian dynamics to improve long-range information propagation and stability in physical modeling problems. By integrating warmup initialization, geometric encoding, and a multi-step training objective, IGNS learns to match the solution trajectories of its underlying port-Hamiltonian dynamics to those of ground-truth PDE solvers, keeping the predicted states on-trajectory and thereby mitigating error accumulation while enhancing long-range dependencies in both spatial and temporal domains. IGNS achieves consistent improvements over strong baselines across six tasks spanning both solid and fluid mechanics, including new benchmarks designed to test complex and long-range dynamics. Our theoretical analysis further shows that the Hamiltonian structure preserves gradient flow, while the port-Hamiltonian extension enables the model to capture a broader class of dynamics through its universality property. Together, these results highlight IGNS as a principled and stable framework for neural simulation of complex physical systems.

**Limitations and Future Work.** Our method currently operates in an open-loop, causal setting, where sources are specified at the start and the model predicts forward without feedback. However, many real-world applications require closed-loop feedback, in which the model continuously adapts its predictions based on observations obtained from sensor measurements or other exogenous inputs. Extending IGNS in this direction is therefore a natural and meaningful next step. Furthermore, IGNS's ability to propagate information over long ranges naturally opens the door to inverse design (Allen et al., 2022) and control tasks (Hoang et al., 2025), where the goal is to identify optimal parameters (e.g., source locations or inputs) that achieve a desired outcome by optimizing inverse objectives. Exploring these directions represents a promising avenue for future work.

ACKNOWLEDGMENTS

This work is part of the M2 subproject of the DFG AI Research Unit 5339, which focuses on learning dynamical process models based on data and expert knowledge to accelerate manufacturing process maturation. We gratefully acknowledge financial support from the German Research Foundation (DFG) and computational resources provided by bwHPC and the HoreKa supercomputer, funded by the Ministry of Science, Research, and the Arts Baden-Württemberg and the German Federal Ministry of Education and Research. We would like to thank Johannes Mitsch and Tobias Würth for developing the data generation platform for the Plate Deformation task in Abaqus. AT, AG, and DB acknowledge funding from EU-EIC EMERGE (Grant No. 101070918).

ETHICS STATEMENT

The research conducted in this paper conforms in every aspect with the ICLR Code of Ethics. Our study does not involve human subjects, sensitive personal data, or applications with foreseeable harmful consequences. No ethical concerns are anticipated regarding data usage, methodology, or findings.

REPRODUCIBILITY STATEMENT

We provide all necessary details to implement our IGNS in Section 5, and describe the choices of hyperparameter for each experiment in Appendix G and full dataset description in Appendix F, thereby ensuring sufficient information to reproduce our results. The full codebase and datasets are released with the camera-ready version to ensure the reproducibility, and are also available at https://thobotics.github.io/neural_pde_matching.

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

## A    LLMs Usage

Large Language Models (LLMs) were used as general-purpose assistive tools to improve the writing quality of this paper. Specifically, we used LLMs to help with grammar correction, rephrasing for clarity, and suggesting some improvements to the overall structure of the text. All LLM-generated text was carefully reviewed and edited by the authors to ensure that it accurately reflects the authors' intentions and scientific content. No LLMs were used to generate scientific content, including but not limited to research direction, hypothesis formulation, experimental design, data analysis, or interpretation of results.

## B    Related Work

**Graph Neural Simulators.**    Graph Neural Networks have been widely adopted in machine learning for physical simulation tasks (Battaglia et al., 2018; Pfaff et al., 2021). Notable examples include modeling particle interactions (Sanchez-Gonzalez et al., 2020; Li et al., 2019c; Sanchez-Gonzalez et al., 2020), fluid dynamics (Pfaff et al., 2021; Yu et al., 2025), and deformable materials (Yu et al., 2024; Linkerhägner et al., 2023). Here, given the observations at the current time step, one can predict the next states by applying a few iterations of message-passing steps to incorporate neighborhood information, and output the next state. This underlying message-passing step is often implemented with graph convolutional layers (Kipf & Welling, 2017) or attention-based methods (Veličković et al., 2018), or more generally, message-passing framework (Battaglia et al., 2018). During rollout, the model can generate the whole trajectories in an autoregressive manner (Pfaff et al., 2021; Wei & Fink, 2025). However, these models often struggle with long-term accuracy due to error accumulation over long time horizons. In contrast, another line of work focuses on directly learning the underlying continuous dynamics using neural ODEs (Chen et al., 2018), which helps mitigate the error accumulation problem by enabling the model to output whole trajectories in a single forward pass and better capture the continuous nature of physical systems. Recent works extend this idea for graph-based architectures (Iakovlev et al., 2021; Lienen & Günnemann, 2022). These models are often trained with multi-step objectives, showing accuracy improvement and with the additional ability to handle irregular time series and improve generalization. However, these models are often limited to short-term predictions (Lienen & Günnemann, 2022; Janny et al., 2023) or supervision in a reduced latent space (Han et al., 2022) due to the challenges in propagating information over both long spatial and temporal horizons.

**Effective propagation in Graph Neural Networks.**    Effectively propagating information across the graph, and thus effectively modeling long-range dependencies, remain a critical challenge for GNNs (Shi et al., 2023a; Rusch et al., 2023; Roth & Liebig, 2024). Stacking many GNN layers to capture long-range dependencies often leads to issues such as over-smoothing (Cai & Wang, 2020; Rusch et al., 2023), when node states become indistinguishable as the number of layers increases. To address this, methods like residual connections (Li et al., 2019a), normalization techniques (Luo et al., 2024), and advanced architectures (Rusch et al., 2022; Eliasof et al., 2021) have been proposed. Another issue that prevent effective long-range propagation is the over-squashing phenomenon (Alon & Yahav, 2021; Topping et al., 2022; Di Giovanni et al., 2023), which arises from topological bottlenecks that squash exponentially increasing amounts of information into node states, causing loss of information between distant nodes. To overcome this, a variety of strategies have been proposed. For example, graph rewiring methods (Topping et al., 2022; Gasteiger et al., 2019; Barbero et al., 2024) modify the graph topology to facilitate communication. In physical system simulation, hierarchical (Deng et al., 2025; Cao et al., 2023; Janny et al., 2023) and rewiring methods, including HCMT (Yu et al., 2024) and PIORF (Yu et al., 2025), improve the global information flow by introducing shortcuts or multi-scale pooling. However, these approaches can make models overly tied to the mesh structure, which can result in geometric overfitting. More recently, Arroyo et al. (2025) demonstrated that both over-smoothing and over-squashing problems are closely linked to the problem of vanishing gradients in message passing, and works like Eliasof et al. (2021); Gravina et al. (2023; 2024a; 2025); Heilig et al. (2025); Arroyo et al. (2025); Trenta et al. (2025); Hariri et al. (2025); Eliasof et al. (2025) have explored this perspective to design GNNs that preserve the gradient flow over many layers. However, most of these methods focus on static graphs and node classification tasks, and it is still unclear how to extend them to physical simulation tasks.

**Neural ODEs and Hamiltonian Dynamics.** Neural Ordinary Differential Equations (ODEs) have been widely used to model continuous-time dynamics (Chen et al., 2018), offering flexibility in time integration and improved generalization, and have also been extended to graph-structured data Poli et al. (2019); Gravina et al. (2024b;c). There has been significant interest in extending neural ODEs to capture physical systems by drawing inspiration from Hamiltonian dynamics (Arnol'd, 2013). Most closely related to our work is HOGN (Greydanus et al., 2019), which parameterizes Hamiltonian dynamics with graph-based message passing, but does not explicitly study long-range spatio-temporal dependencies. Other notable examples include Hamiltonian Neural Networks (HNNs) (Greydanus et al., 2019) and Symplectic ODE-Net (Zhong et al., 2020), which achieve energy conservation and stable long-term predictions by learning a parameterized Hamiltonian function. This conservation property has also been explored as a way to mitigate gradient vanishing in deep neural networks (Galimberti et al., 2023; Lanthaler et al., 2023; Rusch & Rus, 2025), and it has inspired new architectures for enhancing long-range propagation in GNNs (Heilig et al., 2025) as well as long-horizon forecasting in state-space-model (Rusch & Rus, 2025). However, these works remain different from ours, as they focus either on static graphs or purely temporal sequences.

In this work, we propose a novel GNS that leverages dynamical systems theory to preserve the gradient flow across distant nodes in spatial domains. Trained with multi-step loss, this model can mitigate error accumulation and hence, enable accurate long-term physical simulation.

## C   DERIVATIONS

In this section, we establish the connection between the port-Hamiltonian formalism in Eq. (6) and the PDE wave equation defined on a space–time domain. In particular, we show that both can be expressed in the form of a mass–spring–damper system. These two links make clear that our port–Hamiltonian formalism induces a second–order inductive bias, which can be interpreted as a weak form of an underlying wave equation.

### C.1   IGNS AS MASS-SPRING-DAMPER ODE

To begin with, we first consider the following Hamiltonian function:

$$H(t, \boldsymbol{q}, \boldsymbol{p}) = \frac{1}{2} \boldsymbol{p}^T \boldsymbol{M}^{-1} \boldsymbol{p} + \frac{1}{2} \boldsymbol{q}^T \boldsymbol{K} \boldsymbol{q}$$

where we define the generalized coordinates $\boldsymbol{q}$ and momenta $\boldsymbol{p} = \boldsymbol{M}\dot{\boldsymbol{q}}$, and let $\boldsymbol{x} = [\boldsymbol{q}, \boldsymbol{p}]^T$. In fact, this definition is consistent with the Hamiltonian defined in Eq. (7) if we set $\tilde{\sigma}(x) = \frac{1}{2}x^2$ and $\boldsymbol{W} = \mathrm{diag}\left([\boldsymbol{K}^{1/2}, \boldsymbol{M}^{-1/2}]\right)$ and $\boldsymbol{V} = \boldsymbol{0}$. The update of $\boldsymbol{q}$ and $\boldsymbol{p}$ is then given by:

$$\dot{\boldsymbol{q}}(t) = \nabla_{\boldsymbol{p}} H(t, \boldsymbol{q}, \boldsymbol{p}) = \boldsymbol{M}^{-1} \boldsymbol{p}(t)$$
$$\dot{\boldsymbol{p}}(t) = -\nabla_{\boldsymbol{q}} H(t, \boldsymbol{q}, \boldsymbol{p}) = -\boldsymbol{K} \boldsymbol{q}(t).$$

We now eliminate $\boldsymbol{p}(t)$ by substituting $\boldsymbol{p}(t) = \boldsymbol{M}\,\dot{\boldsymbol{q}}(t)$, and add the damping term $\boldsymbol{D}\,\dot{\boldsymbol{q}}(t)$ and the external force $\boldsymbol{r}(t, \boldsymbol{q})$, yielding:

$$\boldsymbol{M}\,\ddot{\boldsymbol{q}}(t) + \boldsymbol{D}\,\dot{\boldsymbol{q}}(t) + \boldsymbol{K}\,\boldsymbol{q}(t) = \boldsymbol{r}(t, \boldsymbol{q}). \tag{11}$$

This recovers the mass-spring-damper system.

For a more general case, consider a Hamiltonian $H(t, \boldsymbol{q}, \boldsymbol{p}) \in \mathcal{C}^2$ in all arguments. Taking the time derivative of $\dot{\boldsymbol{q}} = \nabla_{\boldsymbol{p}} H(t, \boldsymbol{q}, \boldsymbol{p})$ gives

$$\ddot{\boldsymbol{q}} = \nabla_{\boldsymbol{pp}}^2 H\,\dot{\boldsymbol{p}} + \nabla_{\boldsymbol{pq}}^2 H\,\dot{\boldsymbol{q}} + \partial_t \nabla_{\boldsymbol{p}} H.$$

Assuming $\nabla_{\boldsymbol{pp}}^2 H$ exists and is invertible (which holds for certain activation functions, e.g. $\sigma(\cdot) = \tanh(\cdot)$ and regularity conditions for invertibility), define $\boldsymbol{M}(t, \boldsymbol{q}, \boldsymbol{p}) := \left(\nabla_{\boldsymbol{pp}}^2 H(t, \boldsymbol{q}, \boldsymbol{p})\right)^{-1}$. Solving for $\dot{\boldsymbol{p}}$,

$$\dot{\boldsymbol{p}} = \boldsymbol{M}(t, \boldsymbol{q}, \boldsymbol{p})\left[\ddot{\boldsymbol{q}} - \nabla_{\boldsymbol{pq}}^2 H\,\dot{\boldsymbol{q}} - \partial_t \nabla_{\boldsymbol{p}} H\right]. \tag{12}$$

We now use the Hamiltonian equation for $\dot{\boldsymbol{p}}$ with damping and external forcing from Equation (6)

$$\dot{\boldsymbol{p}} = -\nabla_{\boldsymbol{q}} H - \boldsymbol{D}\,\dot{\boldsymbol{q}} + \boldsymbol{r}(t, \boldsymbol{q}) \tag{13}$$

Substituting Equation (12) into Equation (13) and rearranging, we arrive at the general second-order form:

$$\underbrace{M(t, q, p)\,\ddot{q}}_{\text{mass}} + \underbrace{\left[D - M(t, q, p)\nabla_{pq}^2 H\right]\dot{q}}_{\text{damping}} + \underbrace{\nabla_q H}_{\text{stiffness}} = \underbrace{r(t, q) + M(t, q, p)\partial_t \nabla_p H}_{\text{external force \& residual}}. \quad (14)$$

This analysis demonstrates that, for general Hamiltonians, the resulting second-order dynamics exhibit state- and time-dependent mass, damping, and stiffness terms. In particular, for separable Hamiltonians, we have $\nabla_{pq}^2 H = 0$, and for time-invariant Hamiltonians, $\partial_t \nabla_p H = 0$. In these cases, the dynamics reduce to the form of Eq. (11), with $M(t, q, p) = (\nabla_{pp}^2 H)^{-1}$ and $K(t, q, p)q(t) = \nabla_q H(t, q, p)$.

**Comparison with GraphCON.** We can further observe that the baseline GraphCON (Rusch et al., 2022) is in fact a simpler instantiation of Eq. (11). Its update takes the form

$$\ddot{q}(t) + \alpha\dot{q}(t) + \gamma q(t) = \sigma(R_\theta(t, q)),$$

where $R_\theta$ is implemented as a standard graph convolution or attention operator. This corresponds directly to a mass–spring–damper system with $M = I$, $D = \alpha I$, $K = \gamma I$, and external forcing $r(\cdot; G) = \sigma(R_\theta(t, q))$. Notably, the mass matrix $M$ is the identity, meaning that coupling between nodes occurs only through the external forcing term $r$, whereas in our formulation coupling also appears on the left-hand side through $M$, $D$, and $K$.

## C.2 Mass–spring–damper ODE as a second–order PDE

In this subsection, we show how a second-order PDE with damping and forcing reduces to a finite-dimensional mass-spring-damper ODE via Galerkin projection. Interested readers are referred to Igel (2016) for further details.

Let $\Omega \subset \mathbb{R}^d$ be a bounded domain and consider the following damped, forced wave equation

$$\partial_{tt}u + \mathcal{B}\,\partial_t u + \mathcal{L}\,u = g(x, t), \qquad (x, t) \in \Omega \times (0, T], \quad (15)$$

where the solution is $u : \Omega \times (0, T] \to \mathbb{R}$, i.e., a scalar field. We set $\mathcal{L} = -c^2\Delta$ with constant wave speed $c > 0$, where $\Delta$ is the spatial Laplacian, and define $\mathcal{B}$ as a damping operator (e.g., $\alpha I$) acting on the time-derivative term $\partial_t u$.

Let $V \subset H^1(\Omega)$ be the test space. To obtain the weak form of Eq. (15), we take a test function $v \in V$, multiply the strong equation by $v$, and integrate over $\Omega$:

$$\int_\Omega \partial_{tt}u\,v\,dx \;+\; \int_\Omega \mathcal{B}\,\partial_t u\,v\,dx \;-\; c^2\int_\Omega \Delta u\,v\,dx = \int_\Omega g(\cdot, t)\,v\,dx.$$

The first two terms already appear in weak form. For the Laplacian term, we apply integration by parts and assume that $u(t)$ vanishes at the boundary. This gives the following bilinear form:

$$a(u, v) := -c^2\int_\Omega \Delta u\,v\,dx = c^2\int_\Omega \nabla u \cdot \nabla v\,dx.$$

Altogether, the weak form reads: for all $v \in V$,

$$\langle\partial_{tt}u, v\rangle + \langle\mathcal{B}\,\partial_t u, v\rangle + a(u, v) = \langle g(\cdot, t), v\rangle, \quad (16)$$

where $\langle f, g\rangle = \int_\Omega f(x)\,g(x)\,dx$ denotes the $L^2(\Omega)$ inner product.

Choose a finite-dimensional subspace $V_h = \text{span}\{\phi_1, \ldots, \phi_k\} \subset V$ and rewrite the solution as

$$u_h(x, t) = \sum_{j=1}^k q_j(t)\phi_j(x), \quad (17)$$

where $q_j(t)$ are time-dependent coefficients to be determined. Since Eq. (16) holds for all $v = \phi_i$, $i = 1, \ldots, k$, rewriting it using the Eq. (17) yields

$$\sum_{j=1}^k \langle\phi_j, \phi_i\rangle\,\ddot{q}_j(t) + \langle\mathcal{B}\,\phi_j, \phi_i\rangle\,\dot{q}_j(t) + a(\phi_j, \phi_i)\,q_j(t) = \langle g(\cdot, t), \phi_i\rangle. \quad (18)$$

To show the equivalent mass-spring-damper form, we now define the following matrices and vectors:

$$\boldsymbol{M}_{ij} := \langle \phi_j, \phi_i \rangle, \qquad \text{(mass matrix)}$$
$$\boldsymbol{D}_{ij} := \langle \mathcal{B} \phi_j, \phi_i \rangle, \qquad \text{(damping matrix)}$$
$$\boldsymbol{K}_{ij} := a(\phi_j, \phi_i), \qquad \text{(stiffness matrix)}$$
$$\boldsymbol{r}_i(\cdot, t) := \langle g(\cdot, t), \phi_i \rangle, \qquad \text{(external forcing)}$$

with vector $\boldsymbol{q} = [q_j]_{j=1}^k$, thus Eq. (18) becomes

$$\boldsymbol{M} \, \ddot{\boldsymbol{q}}(t) + \boldsymbol{D} \, \dot{\boldsymbol{q}}(t) + \boldsymbol{K} \, \boldsymbol{q}(t) = \boldsymbol{r}(\cdot, t).$$

**Interpretation.** Because our latent dynamics (Eqs. (6) and (14)) have the same form as the system above, IGNS can be seen as learning *effective* mass, damping, and stiffness operators, consistent with a weak-form second-order PDE (in a P1 nodal basis, coefficients equal nodal values, i.e., $u(\boldsymbol{x}_j, t) = \boldsymbol{q}_j(t)$). This method-of-lines view is related to Finite Element Networks (FEN) Lienen & Günnemann (2022), which were the first to introduce the connection between message passing and the finite element method. However, while FEN focuses on *diffusion/heat* dynamics (parabolic, first order in time), here we model a more general *wave* equation with forcing and damping, yielding a more general second-order hyperbolic PDE.

## D  TIME-VARYING WEIGHT MATRICES

To further enhance the model's expressiveness, we propose the following parameterization to make $\boldsymbol{W}(t)$ and $\boldsymbol{V}(t)$ in Eq. (7) time-varying, allowing the Hamiltonian to adapt over time, which is particularly useful for modeling more complex, non-stationary dynamics.

Here, we consider only the case of $\boldsymbol{W}(t)$, as $\boldsymbol{V}(t)$ can be treated similarly. Assuming $\boldsymbol{W}(t)$ is a square matrix of size $d \times d$, we decompose it into symmetric and skew-symmetric parts, i.e., $\boldsymbol{W}(t) = \boldsymbol{S}(t) + \boldsymbol{A}(t)$, where

$$\boldsymbol{S} = \begin{bmatrix} s_{11} & s_{12} & \cdots & s_{1d} \\ s_{12} & s_{22} & \cdots & s_{2d} \\ \vdots & \vdots & \ddots & \vdots \\ s_{1d} & s_{2d} & \cdots & s_{dd} \end{bmatrix}, \qquad \boldsymbol{A} = \begin{bmatrix} 0 & a_{12} & \cdots & a_{1d} \\ -a_{12} & 0 & \cdots & a_{2d} \\ \vdots & \vdots & \ddots & \vdots \\ -a_{1d} & -a_{2d} & \cdots & 0 \end{bmatrix},$$

with $s_{ij} = s_{ji}$, $a_{ij} = -a_{ji}$, and $a_{ii} = 0$. We then factor these parts through time-invariant and time-varying components as

$$\boldsymbol{S}(t) = \boldsymbol{U}_s \, \mathrm{diag}(\gamma_\theta(t)) \, \boldsymbol{U}_s^\top, \qquad \boldsymbol{A}(t) = \boldsymbol{U}_a \, \mathrm{diag}(\tau_\theta(t)) \, \boldsymbol{P}_a^\top - \boldsymbol{P}_a \, \mathrm{diag}(\tau_\theta(t)) \, \boldsymbol{U}_a^\top,$$

where $\boldsymbol{U}_s, \boldsymbol{U}_a, \boldsymbol{P}_a$ are learned (time-invariant) bases, and $\gamma_\theta(t), \tau_\theta(t)$ are two time-varying coefficient vectors output by MLPs with parameters $\theta$. This two-step construction helps keeping all the learned parameters unconstrained while reducing the parameter counts to $\mathcal{O}(d)$, compared to the naive approach of letting a MLP output all the matrix entries, which leads to $d^2$ parameters. In practice, we implement $\gamma_\theta(t), \tau_\theta(t)$ as small MLPs that take time-embedding $t$ as input and output the corresponding coefficient vectors.

## E  PROOFS OF THE THEORETICAL STATEMENTS AND FURTHER DISCUSSION

In this section, we prove the theoretical statements in this paper together with a more in-depth discussion.

### E.1  NEURAL OSCILLATORS AND UNIVERSALITY

We start this section by recalling the definition of neural oscillators and their universality property (Lanthaler et al., 2023). The general setting involves learning an operator $\Phi : \boldsymbol{u} \to \Phi(\boldsymbol{u})$, that maps a time-dependent input signal $\boldsymbol{u}(t)$ to an output function $\Phi(\boldsymbol{u})(t) \in \mathbb{R}^q$. Here, $\Phi$ is defined as a continuous operator (w.r.t. the $L^\infty$ norm) that maps the space

$$C_0([0, T]; \mathbb{R}^p) = \{\boldsymbol{u} : [0, T] \to \mathbb{R}^p | t \mapsto u(t) \text{ is continuous and } \boldsymbol{u}(0) = 0\} \tag{19}$$

into itself. The assumption $\boldsymbol{u}(0) = 0$ is not restrictive, as the main paper showed that the proposed neural oscillators can operate even in the case $\boldsymbol{u}(0)$ is non-zero. Here, by adding an arbitrarily small time interval to "warmup", the oscillator starting from the rest state can synchronize with this non-zero input signal. The only remaining assumption is that $\Phi$ needs to be *causal*, in the sense that the value of $\Phi(u)(t)$ does not depend on future values $\{u(\tau)|\tau > t\}$, which matches our setup considered in this paper well. Formally speaking, $\Phi$ is causal if giving two input signals and assuming that $u|_{[0,t]} = v|_{[0,t]}$, then $\Phi(u)(t) = \Phi(v)(t)$.

Then, we recall the general form of the neural oscillator, given in the following

$$\begin{cases} \ddot{\boldsymbol{y}}(t) = \sigma\left(\boldsymbol{A}\boldsymbol{y}(t) + \boldsymbol{B}\boldsymbol{u}(t) + \boldsymbol{c}\right), \\ \boldsymbol{y}(0) = \dot{\boldsymbol{y}}(0) = \boldsymbol{0}, \\ \boldsymbol{z}(t) = \boldsymbol{D}\boldsymbol{y}(t) + \boldsymbol{e}, \end{cases} \tag{20}$$

where $\boldsymbol{y} \in \mathbb{R}^d$ and $\sigma$ is a $C^{\infty}(\mathbb{R})$ activation function with $\sigma(0) = 0$ and $\sigma'(0) = 1$, e.g., $\tanh$ or $\sin$. Here, Eq. (20) defines an input-output mapping $\boldsymbol{u}(t) \mapsto \boldsymbol{z}(t) \in \mathbb{R}^q$ as a solution of a second-order dynamics. Furthermore, Lanthaler et al. (2023) (Section 2) also considers another form of neural oscillator, which is given by

$$\begin{cases} \ddot{\boldsymbol{y}}(t) = \sigma\left(\boldsymbol{A}\boldsymbol{y}(t) + \boldsymbol{c}\right) + \boldsymbol{r}(t), \\ \boldsymbol{y}(0) = \dot{\boldsymbol{y}}(0) = \boldsymbol{0}, \\ \boldsymbol{z}(t) = \boldsymbol{D}\boldsymbol{y}(t) + \boldsymbol{e}. \end{cases} \tag{21}$$

In this case, the input is provided via $\boldsymbol{r}(t)$, an external forcing term which in their setting is a linear transformation of the inputs. For the universality result presented shortly below, we will consider the neural oscillator in Eq. (21), which is the closest setting to ours. Here, the universality result remains the same, as the original proof does not change if the forcing is inside or outside the activation (see the discussion at the end of Section 2 in Lanthaler et al. (2023)).

The main result of Lanthaler et al. (2023), for Neural Oscillators in the form of Equation (21), can be summarized in the following statement:

**Theorem 3** (Theorem 3.1 of Lanthaler et al. (2023)). *Let $\Phi : C_0([0,T]; \mathbb{R}^p) \to C_0([0,T]; \mathbb{R}^p)$ be a causal and continuous operator and let $K \subset C_0([0,T]; \mathbb{R}^p)$ be compact. Then, for any $\epsilon > 0$, there exists a latent dimension $d$, weights $\boldsymbol{A}, \boldsymbol{D}$, and biases $\boldsymbol{c}, \boldsymbol{e}$ such that the output of the multi-layer neural oscillator with input satisfies:*

$$\sup_{t \in [0,T]} |\Phi(u)(t) - z(t)| \le \epsilon, \quad \forall u \in K. \tag{22}$$

While this theorem is stated for maps between time-dependent signals $\boldsymbol{u}(t)$ and $\boldsymbol{z}(t)$, Lanthaler et al. (2023) provides an additional result for continuous functions $F : \mathbb{R}^p \to \mathbb{R}^q$, i.e., neural oscillators can learn to approximate to arbitrary precision any such map $F$ between *vector spaces*. The full discussion is provided in Appendix A of Lanthaler et al. (2023).

To show that our model is universal in either sense, we need to cast our model, defined by Eq. (6) and Eq. (7), into the definition in Eq. (21). First, we provide a proof of the universality of our model, cast into the neural oscillator form, and then we provide an interpretation of our results.

**Theorem** (Universality of IGNS, Theorem 1 of Section 4). *Let $\Psi_\theta$ be the functional that maps the initial condition $\boldsymbol{x}_0 = [\boldsymbol{q}_0, \boldsymbol{p}_0]$ to the solution at time $\tau$ to the ODE defined Eq. (6) and Eq. (7). In other words, $\Psi_\theta$ represents IGNS. Then, for any $F : \mathbb{R}^{n \times d} \to \mathbb{R}^{n \times d}$ with compact support, which represents the evolution of the system after time $\tau$, there exist a set of parameters $\theta$ such that $\|\Psi_\theta(\boldsymbol{x}_0) - F(\boldsymbol{x}_0)\| \le \epsilon$.*

*Proof.* Our goal for the proof is to reconcile our definition of the port-Hamiltonian system in Eq. (6) with the definition of the oscillatory model in Eq. (21). We start from the derivation of IGNS as a mass-spring-damper system in Eq. (14). Here, we consider the separable Hamiltonian as in Eq. (7), hence $\nabla_{pq}^2 H = 0$. We can additionally set the dissipation term to be null $\boldsymbol{D} = 0$: as we will see, without this term, our model can be cast more simply into the form of the Neural Oscillator in Equation (21). Adding this term allows for further flexibility to learn a wider class of functionals, but does not break the proof ($\boldsymbol{D}$ can be inserted into the definition of the auxiliary function we use

later). Conversely, IGNS can simply learn to have a null dissipation term, if needed by the particular task. With this in mind, Eq. (14) can be written as

$$\ddot{\boldsymbol{q}} = -\boldsymbol{M}^{-1} \left( \nabla_{\boldsymbol{q}} H + \boldsymbol{r}(t, \boldsymbol{q}) \right) + \partial_t \nabla_{\boldsymbol{p}} H. \tag{23}$$

For this proof, we consider the time-invariant case, while the general case is very similar, as $\partial_t \nabla_{\boldsymbol{p}} H$ plays the same role as the external forcing $\boldsymbol{r}(t, \boldsymbol{q})$. By using the definition of the Hamiltonian in Eq. (7), we rewrite Eq. (23) for each node as

$$\ddot{\boldsymbol{q}}_i = -\boldsymbol{M}^{-1} \left( \boldsymbol{W_q} \sigma \left( \boldsymbol{W_q} \boldsymbol{q}_i + \sum_{j \in \mathcal{N}(i)} \boldsymbol{V_q} \boldsymbol{q}_j + \boldsymbol{b} \right) + \boldsymbol{r}(t, \boldsymbol{q}) + \boldsymbol{g}(\boldsymbol{q})_i \right) \tag{24}$$

where $\boldsymbol{g}(\boldsymbol{q})_i = \sum_{j \in \mathcal{N}(i)} \boldsymbol{V_q} \sigma(\boldsymbol{W_q} \boldsymbol{q}_j + \sum_{k \in \mathcal{N}(j)} \boldsymbol{V_q} \boldsymbol{q}_k + \boldsymbol{b})$ accounts for the terms in which node $i$ is a neighbor of node $j$ (there exists a $k \in \mathcal{N}(j)$ for which $k = i$). Stacking all nodes, we arrive at the form

$$\ddot{\boldsymbol{q}} = -\boldsymbol{M}^{-1} \left( \tilde{\boldsymbol{W}}_{\boldsymbol{q}} \sigma \left( \tilde{\boldsymbol{W}}_{\boldsymbol{q}} \boldsymbol{q} + \tilde{\boldsymbol{b}} \right) + \tilde{\boldsymbol{r}}(t, \boldsymbol{q}) \right), \tag{25}$$

where $\tilde{\boldsymbol{b}} = [\boldsymbol{b}, \dots, \boldsymbol{b}]^\top$ contains one copy of $\boldsymbol{b}$ for each node and $\tilde{\boldsymbol{W}}$ is a block matrix, where each diagonal block has a copy of $\boldsymbol{W}$ and block $i, j$ has a copy of $\boldsymbol{V}$ iff $j \in \mathcal{N}(i)$. In fact, $\tilde{\boldsymbol{W}}$ absorbs the effects of node aggregation into one bigger block matrix.

Since the definition from Lanthaler et al. (2023) has a null initial condition, we adopt a trick with the forcing terms to reconcile the two forms finally. Let $\bar{\boldsymbol{q}}, \bar{\boldsymbol{p}}$ be auxiliary functions, solutions to $\dot{\boldsymbol{q}} = \boldsymbol{p}$ and $\dot{\boldsymbol{p}} = \boldsymbol{M}^{-1}(\boldsymbol{q}, \boldsymbol{p})\tilde{\boldsymbol{r}}(t, \boldsymbol{q})$, which lead to $\ddot{\boldsymbol{q}} = \boldsymbol{M}^{-1}(\boldsymbol{q}, \boldsymbol{p})\tilde{\boldsymbol{r}}(t, \boldsymbol{q})$, and initial conditions $\boldsymbol{q}_0$ and $\boldsymbol{p}_0$. Then, the function $\boldsymbol{y} = \boldsymbol{q} - \bar{\boldsymbol{q}}$ evolves through the following equations

$$\begin{aligned} \ddot{\boldsymbol{y}} &= -\boldsymbol{M}^{-1}(\boldsymbol{y} + \bar{\boldsymbol{q}}, \dot{\boldsymbol{y}} + \bar{\boldsymbol{p}}) \tilde{\boldsymbol{W}}_{\boldsymbol{q}} \sigma \left( \tilde{\boldsymbol{W}}_{\boldsymbol{q}} \boldsymbol{y} + \tilde{\boldsymbol{W}}_{\boldsymbol{q}} \bar{\boldsymbol{q}} + \tilde{\boldsymbol{b}} \right) \\ &\quad - \boldsymbol{M}^{-1}(\boldsymbol{y} + \bar{\boldsymbol{q}}, \dot{\boldsymbol{y}} + \bar{\boldsymbol{p}}) \tilde{\boldsymbol{r}}(t, \boldsymbol{y} + \bar{\boldsymbol{q}}) - \boldsymbol{M}^{-1}(\bar{\boldsymbol{q}}, \bar{\boldsymbol{p}}) \tilde{\boldsymbol{r}}(t, \bar{\boldsymbol{q}}), \\ \boldsymbol{y}(0) &= \boldsymbol{0}, \\ \dot{\boldsymbol{y}}(0) &= \boldsymbol{0}. \end{aligned} \tag{26}$$

Here, we can see that Eq. (26) is indeed in a similar form as Eq. (21). The only difference is in the additional matrix $\boldsymbol{M}^{-1} = \nabla^2_{pp} H$, which depends only on $\boldsymbol{W_p}$ (since $H$ is separable in $\boldsymbol{p}, \boldsymbol{q}$), and makes Eq. (26) more general than Eq. (21). Finally, we define $\boldsymbol{z}(t)$ as either $\boldsymbol{z}(t) = \boldsymbol{y}(t)$ or as obtained via a decoder $\boldsymbol{z}(t) = \mathrm{dec}(\boldsymbol{y}(t))$. Hence, IGNS is a universal neural oscillator by Theorem 3. To conclude our proof, we need two extra observations. First, since $\boldsymbol{p}(t) = \boldsymbol{M}\dot{\boldsymbol{q}}(t)$, IGNS is universal also in terms of $\boldsymbol{x} = [\boldsymbol{q}, \boldsymbol{p}]^T$. Second, this result shows that IGNS is an operator that can convert an input signal $\boldsymbol{r}(t, \boldsymbol{q})$ into any possible evolution, and it can be further extended to any map between the space of our interest $\mathbb{R}^{n \times d}$ to itself. For this, it is sufficient to consider the proof given in Appendix A of Lanthaler et al. (2023) and recall the definition of $\bar{\boldsymbol{q}}$, which adds the information on the initial condition through $\tilde{\boldsymbol{r}}(t, \bar{\boldsymbol{q}})$. □

**Remark 2.** *This proof also shows that IGNS can learn any possible transformation with compact support between the forcing terms and the dynamics of the physical system. Specifically, the Hamiltonian in Eq. (7) can learn any transformation between continuous functions in $C([0, T], \mathbb{R}^{n \times d})$.*

**Remark 3.** *Like most Universality theorems for Neural networks or operators, our result requires the support of the operator $F$ to be a compact subspace of $\mathbb{R}^{n \times d}$. We argue that this assumption is not restrictive for our purposes, as we work with physical simulations. In particular, a subset $K \subset \mathbb{R}^{n \times d}$ is compact if it is closed and bounded. In our case, this means that there exists $A > 0$ such that $\|\boldsymbol{q}_0\| \leq A$ and $\|\boldsymbol{p}_0\| \leq A$. It is reasonable to assume that in contained physical simulations, the scale of physical quantities is limited and that there is a limit to the energy inside or added to the system. Hence, these are fair assumptions for our purposes.*

There is an interesting interpretation of this result, which links the definition of IGNS in Eq. (6) and the one of neural oscillators. In particular, looking at the formulation in Eq. (26), we see that all the terms that are outside the activation function play the same role as the input signal $\boldsymbol{r}(t)$, defined in Eq. (21). In our case, these terms contain the geometrical information and the external or driving forces acting on the system, as well as the initial condition. Hence, IGNS learns to transform these

inputs into the desired dynamics. In this sense, the Hamiltonian learns how to propagate through the mesh the information from the initial conditions, geometry, and external forcing.

Another interpretation comes directly from Eq. (24) by looking at the state of each node $\boldsymbol{q}_i$. In particular, we can interpret the state of neighboring nodes $\boldsymbol{q}_j$, with $j \in \mathcal{N}(i)$, as the input function $\boldsymbol{u}(t)$. Hence, IGNS learns how each node should behave in the simulation as a consequence of the influence of the neighboring nodes, dissipation, and forcing.

### E.2 Vanishing Gradients and Long-range Interactions

The radius of propagation exploited by the GNN has a large influence on its performance (Li et al., 2019a; Cai & Wang, 2020; Alon & Yahav, 2021; Rusch et al., 2023; Di Giovanni et al., 2023; Roth & Liebig, 2024), as it is often related to over-smoothing, which causes node representations collapse, and over-squashing, which causes information dissipation. Recently, it has been observed that these behaviors are strongly related to the sensitivity of the Jacobians $\partial \boldsymbol{x}^{(l+1)}/\partial \boldsymbol{x}^{(l)}$ (or $\partial \boldsymbol{x}_i^{(l)}/\partial \boldsymbol{x}_j^{(0)}$) of the GNN updates and their norms (Arroyo et al., 2025), as low sensitivity values can cause a contractive behavior that induces information loss. Hence, it is necessary to employ models that provide non-contractive updates to avoid these issues. Additionally, real-world simulations often require modeling long-range interactions, as physical information can propagate over long distances. In this discussion, we aim to show that all these problems can be solved or limited by adopting an oscillatory dynamic at the core of the model. Particularly, the Hamiltonian formalism provides further guarantees on vanishing gradients (Galimberti et al., 2021; 2023) and long-range interactions (Heilig et al., 2025), strengthening performance. From a dynamical system perspective, a key aspect of oscillatory systems lies in their purely imaginary eigenvalues:

**Theorem 4** (Eigenvalues of an Oscillatory System). *Consider a second-order system of the form $\boldsymbol{M}\ddot{\boldsymbol{x}} + \boldsymbol{K}\boldsymbol{x} = 0$, where $\boldsymbol{M}$ and $\boldsymbol{K}$ are positive definite. Then, the eigenvalues of the system are pure imaginary.*

*Proof.* Since $\boldsymbol{M}$ is positive definite, it is invertible, and we can consider the following system instead:

$$\ddot{\boldsymbol{x}} + \boldsymbol{K}\boldsymbol{x} = 0. \tag{27}$$

We decompose this second-order system into a system of first-order ODEs:

$$\begin{aligned}
\dot{\boldsymbol{x}} &= \boldsymbol{v}, \\
\dot{\boldsymbol{v}} &= -\boldsymbol{K}\boldsymbol{x}.
\end{aligned} \tag{28}$$

Calling $\boldsymbol{y} = [\boldsymbol{x}, \boldsymbol{v}]^\top$, the system can be represented as

$$\dot{\boldsymbol{y}} = \begin{bmatrix} \boldsymbol{0} & \boldsymbol{I} \\ -\boldsymbol{K} & \boldsymbol{0} \end{bmatrix} \boldsymbol{y} = \begin{bmatrix} \boldsymbol{0} & \boldsymbol{I} \\ -\boldsymbol{I} & \boldsymbol{0} \end{bmatrix} \begin{bmatrix} \boldsymbol{K} & \boldsymbol{0} \\ \boldsymbol{0} & \boldsymbol{I} \end{bmatrix} \boldsymbol{y}. \tag{29}$$

That is, the evolution is determined by a matrix which is the product of an antisymmetric matrix and a positive definite one. Hence, as per Lemma 2 of Galimberti et al. (2021), the eigenvalues of the system are all imaginary. $\qquad\square$

In terms of sensitivity matrices, a similar property has been derived for general model dynamics defined by $\dot{\boldsymbol{x}} = \boldsymbol{f}(\boldsymbol{x}(t))$ (Heilig et al., 2025; Galimberti et al., 2023; 2021). The starting point is understanding how the sensitivity matrix $\partial \boldsymbol{x}(T)/\partial \boldsymbol{x}(T-t)$ evolves through time, measuring the effect of past states on the current one:

**Lemma 1** (Lemma 1 of Galimberti et al. (2021)). *Let $\boldsymbol{x}$ be the solution to the first-order ODE $\dot{\boldsymbol{x}} = \boldsymbol{f}(\boldsymbol{x}(t))$. Then, calling $\phi(T, T-t) = \frac{\partial \boldsymbol{x}(T)}{\partial \boldsymbol{x}(T-t)}$, it holds that*

$$\frac{\mathrm{d}}{\mathrm{d}t} \phi(T, T-t) = \frac{\partial \boldsymbol{f}}{\partial \boldsymbol{x}}\bigg|_{\boldsymbol{x}(T-t)} \phi(T, T-t) \tag{30}$$

Hence, the sensitivity matrix and, most importantly, the vanishing gradient effect are strictly related to the eigenvalues of $\partial \boldsymbol{f}/\partial \boldsymbol{x}$ which, for oscillatory models, are purely imaginary. In reality, Heilig et al. (2025) shows a much stronger property for IGNS$_{\textbf{ti}}$. In particular:

**Theorem** (Theorem 2 of Section 4 and Theorem 2.3 of Heilig et al. (2025)). *If $\boldsymbol{x}(t)$ is the solution to the ODE $\dot{\boldsymbol{x}}_i = J\nabla_{\boldsymbol{x}_i} H_\theta(t, \boldsymbol{X})$, that is, the Hamiltonian dynamic with Eq.* (6) *without dampening and external forcing, then $\left\| \frac{\partial \boldsymbol{x}(t)}{\partial \boldsymbol{x}(s)} \right\| \geq 1$ for any $0 \leq s \leq t$.*

This result represents a lower bound on the sensitivity matrix norm for any time $t$. This ensures that gradients never vanish, even over extended periods. We emphasize that not all oscillatory dynamics exhibit the same desirable properties as IGNS. For instance, while GraphCON (Rusch et al., 2022) models a mass-spring-damper system (see Section C.1) and alleviates some of the propagation issues presented. A Hamiltonian-like dynamic (Heilig et al., 2025), as adopted in our IGNS, provides stronger theoretical guarantees on long-range interactions and sensitivity matrices. These advantages also translate into improved empirical performance, as demonstrated in Section 5. GraphCON adds node coupling only in its forcing term, while employing diagonal matrices in the left-hand side of Eq. (11). See Section C.1 for additional details on the interpretation of these models as mass-spring-damper systems.

**First-order models and the Heat Equation.** Without inductive biases introduced in oscillatory and second-order models, there is no theoretical guarantee of preventing vanishing gradients. On the contrary, there are both practical (Li et al., 2019c) and theoretical (Arroyo et al., 2025) insights that show how GNNs are prone to these exponentially decaying gradients. Similarly, first-order differential-equation-inspired GNNs are based on the heat equation $\partial_t u - \Delta u = 0$, which leads again to exponentially decaying gradient and information propagation (Gravina et al., 2025), provable also through the lenses of over-smoothing (Eliasof et al., 2021).

### E.3 FURTHER DETAILS ON THE VANISHING GRADIENTS WITH DISSIPATION AND EXTERNAL FORCING

In this Section, we provide further details on how IGNS limits the vanishing gradient effect. This discussion is inspired by the results in Rusch et al. (2022), in particular Propositions 3.5 and 3.6. We formalize this in the following statement:

**Theorem 5.** *For $t = 1, \ldots, T$, let $\boldsymbol{x}^{(t)}$ the prediction of IGNS at time $t$. Then, for sufficiently small $\Delta t << 1$, the leading order of $\Delta t$ in the gradient of the multi-step loss term at time $T$, with respect to a parameter at time $t$, $\boldsymbol{w}^{(t)}$ (that is $\frac{\partial \mathcal{L}^{(T)}}{\partial \boldsymbol{w}^{(t)}}$), is independent of the number of iterations. Thus, although it can be small, the gradient will not vanish even with many iterations of the model.*

*Proof.* We follow the steps for the proof of Proposition 3.6 in Rusch et al. (2022). First, we use the chain rule to calculate the gradient of the loss with respect to the model parameters. We will consider only one term of the multistep loss, the one for the $T$-th step, which we call $\mathcal{L}$ for simplicity. As we will see at the end of the proof, this will not be an issue. We are interested in the influence of a model parameter $\boldsymbol{w}$ at step $s$, which we indicate as $\boldsymbol{w}^{(s)}$, on the loss at step $T$.

$$\frac{\partial \mathcal{L}}{\partial \boldsymbol{w}^t} = \frac{\partial \mathcal{L}}{\partial \boldsymbol{x}^{(T)}} \frac{\partial \boldsymbol{x}^{(T)}}{\partial \boldsymbol{x}^{(t)}} \frac{\partial \boldsymbol{x}^{(t)}}{\partial \boldsymbol{w}^{(s)}} = \frac{\partial \mathcal{L}}{\partial \boldsymbol{x}^{(T)}} \prod_{t=s}^{T} \frac{\partial \boldsymbol{x}^{(t+1)}}{\partial \boldsymbol{x}^{(t)}} \frac{\partial \boldsymbol{x}^{(t)}}{\partial \boldsymbol{w}^{(s)}} \tag{31}$$

Thus, we need to calculate the one-step Jacobian $\frac{\partial \boldsymbol{x}^{(t+1)}}{\partial \boldsymbol{x}^{(t)}}$ for this analysis, which requires calculating the Jacobians of the combinations of the $\boldsymbol{p}$ and $\boldsymbol{q}$ terms. The calculations for the port-Hamiltonian dynamics of IGNS are very similar to those of Heilig et al. (2025). The main differences are that our weight matrices are time-dependent (which does not influence the calculations of partial derivatives with respect to $\boldsymbol{x}^{(t)}$) and that our dissipative term does not depend on $\boldsymbol{q}$. Since we are interested in the leading order terms (as in Rusch et al. (2022)), we report the four quantities of interest up to order $\Delta t$, which we adapt from Heilig et al. (2025) (equations (48) and (49), here we exchange between numerator and denominator notation). We omit the explicit time-dependence of the matrix

and recall it when needed.

$$\frac{\partial \boldsymbol{p}_i^{(t+1)}}{\partial \boldsymbol{p}_j^{(t)}} = \boldsymbol{I}_{ij} - \Delta t \bigg( \boldsymbol{I}_{ij} \boldsymbol{D}_\theta \sigma_p'(i) \boldsymbol{W}_p^2$$

$$+ \boldsymbol{A}_{ij} \boldsymbol{D}_\theta \bigg( \sigma_p'(i) \boldsymbol{V}_p \boldsymbol{W}_p + \sigma_p'(j) \boldsymbol{W}_p \boldsymbol{V}_p + \sum_{k \in \mathcal{N}_i \cap \mathcal{N}_j} \sigma_p'(k) \boldsymbol{V}_p^2 \bigg) \bigg)$$

$$\frac{\partial \boldsymbol{p}_i^{(t+1)}}{\partial \boldsymbol{q}_j^{(t)}} = -\Delta t \bigg( \boldsymbol{I}_{ij} \bigg( \sigma_q'(i) \boldsymbol{W}_q^2 + \sum_{k \in \mathcal{N}_i} \sigma_q'(k) \boldsymbol{V}_q^2 \bigg)$$

$$+ \boldsymbol{A}_{ij} \bigg( \sigma_q'(i) \boldsymbol{V}_q \boldsymbol{W}_q + \sigma_q'(j) \boldsymbol{W}_q \boldsymbol{V}_q + \sum_{k \in \mathcal{N}_i \cap \mathcal{N}_j} \sigma_q'(k) \boldsymbol{V}_q^2 \bigg) + \frac{\partial \boldsymbol{F}(\boldsymbol{q}, t)}{\partial \boldsymbol{q}_j} \bigg) \quad (32)$$

$$\frac{\partial \boldsymbol{q}_i^{(t+1)}}{\partial \boldsymbol{p}_j^{(t)}} = +\Delta t \bigg( \boldsymbol{I}_{ij} \bigg( \sigma_p'(i) \boldsymbol{W}_p^2 + \sum_{k \in \mathcal{N}_i} \sigma_p'(k) \boldsymbol{V}_p^2 \bigg)$$

$$+ \boldsymbol{A}_{ij} \bigg( \sigma_p'(i) \boldsymbol{V}_p \boldsymbol{W}_p + \sigma_p'(j) \boldsymbol{W}_p \boldsymbol{V}_p + \sum_{k \in \mathcal{N}_i \cap \mathcal{N}_j} \sigma_p'(k) \boldsymbol{V}_p^2 \bigg) \bigg)$$

$$\frac{\partial \boldsymbol{q}_i^{(t+1)}}{\partial \boldsymbol{q}_j^{(t)}} = \boldsymbol{I}_{ij} + O(\Delta t^2)$$

where we used the same notation as in Heilig et al. (2025) for $\sigma_p'(i) = \sigma' \left( \boldsymbol{W}_p \boldsymbol{p}_i + \sum_{k \in \mathcal{N}_i} \boldsymbol{V}_p \boldsymbol{p}_k \right)$ and the other similar terms.

Now, we repeat the process of Propositions 3.5 and 3.6 from Rusch et al. (2022) and calculate $\frac{\partial \boldsymbol{x}^{(t+2)}}{\partial \boldsymbol{x}^{(t+1)}} \frac{\partial \boldsymbol{x}^{(t+1)}}{\partial \boldsymbol{x}^{(t)}}$ up to the first-order, to see how the terms in the chain rule multiply. To do so, we need to calculate the four submatrices of this product. Considering that only the first and fourth terms in the above calculations have a leading term of order $O(1)$, we can simplify as

$$\frac{\partial \boldsymbol{p}^{(t+2)}}{\partial \boldsymbol{p}^{(t)}} = \frac{\partial \boldsymbol{p}^{(t+2)}}{\partial \boldsymbol{p}^{(t+1)}} \frac{\partial \boldsymbol{p}^{(t+1)}}{\partial \boldsymbol{p}^{(t)}} + \frac{\partial \boldsymbol{p}^{(t+2)}}{\partial \boldsymbol{q}^{(t+1)}} \frac{\partial \boldsymbol{q}^{(t+1)}}{\partial \boldsymbol{p}^{(t)}} = \frac{\partial \boldsymbol{p}^{(t+2)}}{\partial \boldsymbol{p}^{(t+1)}} + \frac{\partial \boldsymbol{p}^{(t+1)}}{\partial \boldsymbol{p}^{(t)}} - \boldsymbol{I} + O(\Delta t^2)$$

$$\frac{\partial \boldsymbol{p}^{(t+2)}}{\partial \boldsymbol{q}^{(t)}} = \frac{\partial \boldsymbol{p}^{(t+2)}}{\partial \boldsymbol{q}^{(t+1)}} \frac{\partial \boldsymbol{q}^{(t+1)}}{\partial \boldsymbol{q}^{(t)}} + \frac{\partial \boldsymbol{p}^{(t+2)}}{\partial \boldsymbol{p}^{(t+1)}} \frac{\partial \boldsymbol{p}^{(t+1)}}{\partial \boldsymbol{p}^{(t)}} = \frac{\partial \boldsymbol{p}^{(t+2)}}{\partial \boldsymbol{q}^{(t+1)}} + \frac{\partial \boldsymbol{p}^{(t+1)}}{\partial \boldsymbol{q}^{(t)}} + O(\Delta t^2)$$

$$\frac{\partial \boldsymbol{q}^{(t+2)}}{\partial \boldsymbol{p}^{(t)}} = \frac{\partial \boldsymbol{q}^{(t+2)}}{\partial \boldsymbol{q}^{(t+1)}} \frac{\partial \boldsymbol{q}^{(t+1)}}{\partial \boldsymbol{p}^{(t)}} + \frac{\partial \boldsymbol{q}^{(t+2)}}{\partial \boldsymbol{p}^{(t+1)}} \frac{\partial \boldsymbol{p}^{(t+1)}}{\partial \boldsymbol{p}^{(t)}} = \frac{\partial \boldsymbol{q}^{(t+2)}}{\partial \boldsymbol{p}^{(t+1)}} + \frac{\partial \boldsymbol{q}^{(t+1)}}{\partial \boldsymbol{p}^{(t)}} + O(\Delta t^2)$$

$$\frac{\partial \boldsymbol{q}^{(t+2)}}{\partial \boldsymbol{q}^{(t)}} = \frac{\partial \boldsymbol{q}^{(t+2)}}{\partial \boldsymbol{q}^{(t+1)}} \frac{\partial \boldsymbol{q}^{(t+1)}}{\partial \boldsymbol{q}^{(t)}} + \frac{\partial \boldsymbol{q}^{(t+2)}}{\partial \boldsymbol{p}^{(t+1)}} \frac{\partial \boldsymbol{p}^{(t+1)}}{\partial \boldsymbol{q}^{(t)}} = \frac{\partial \boldsymbol{q}^{(t+2)}}{\partial \boldsymbol{q}^{(t+1)}} + \frac{\partial \boldsymbol{q}^{(t+1)}}{\partial \boldsymbol{q}^{(t)}} - \boldsymbol{I} + O(\Delta t^2)$$

(33)

This is very important, as it shows that up to the first order,

$$\frac{\partial \boldsymbol{x}^{(t+2)}}{\partial \boldsymbol{x}^{(t+1)}} \frac{\partial \boldsymbol{x}^{(t+1)}}{\partial \boldsymbol{x}^{(t)}} = \frac{\partial \boldsymbol{x}^{(t+2)}}{\partial \boldsymbol{x}^{(t+1)}} + \frac{\partial \boldsymbol{x}^{(t+1)}}{\partial \boldsymbol{x}^{(t)}} - \boldsymbol{I} + O(\Delta t^2) \quad (34)$$

which is very similar to the results in Rusch et al. (2022). With similar steps, we arrive at

$$\frac{\partial \boldsymbol{x}^{(T)}}{\partial \boldsymbol{x}^{(s)}} = \sum_{t=s}^{T-1} \left( \frac{\partial \boldsymbol{x}^{(t+1)}}{\partial \boldsymbol{x}^{(t)}} - \boldsymbol{I} \right) + O(\Delta t)^2$$

$$= \sum_{t=s}^{T-1} \left( \frac{\partial \boldsymbol{x}^{(t+1)}}{\partial \boldsymbol{x}^{(t)}} \right) - (T - s - 2)\boldsymbol{I} + O(\Delta t^2) \quad (35)$$

$$= \left( \boldsymbol{I} + \sum_{t=s}^{T-1} \boldsymbol{E}^{t+1,t} \right)$$

where, in our case, $\boldsymbol{E}^{t+1,t} = \frac{\partial \boldsymbol{x}^{(t+1)}}{\partial \boldsymbol{x}^{(t)}} - \boldsymbol{I}$ and has leading order $\Delta t$. Finally, we note from Equation (32) that $\left( \frac{\partial \boldsymbol{x}^{(t)}}{\partial \boldsymbol{w}^{(t)}} \right)$, that is, the gradient with respect to parameters, has leading order $O(1)$ or $O(\Delta t)$, and, naturally $\frac{\partial \mathcal{L}}{\partial \boldsymbol{x}^{(T)}}$ is of order $O(1)$. Hence, as in Rusch et al. (2022), the leading order of the gradients $\frac{\partial \mathcal{L}^{(T)}}{\partial \boldsymbol{w}^{(t)}}$ is $O(\Delta t^2)$ independent on the number of iterations between $t$ and $T$.  $\square$

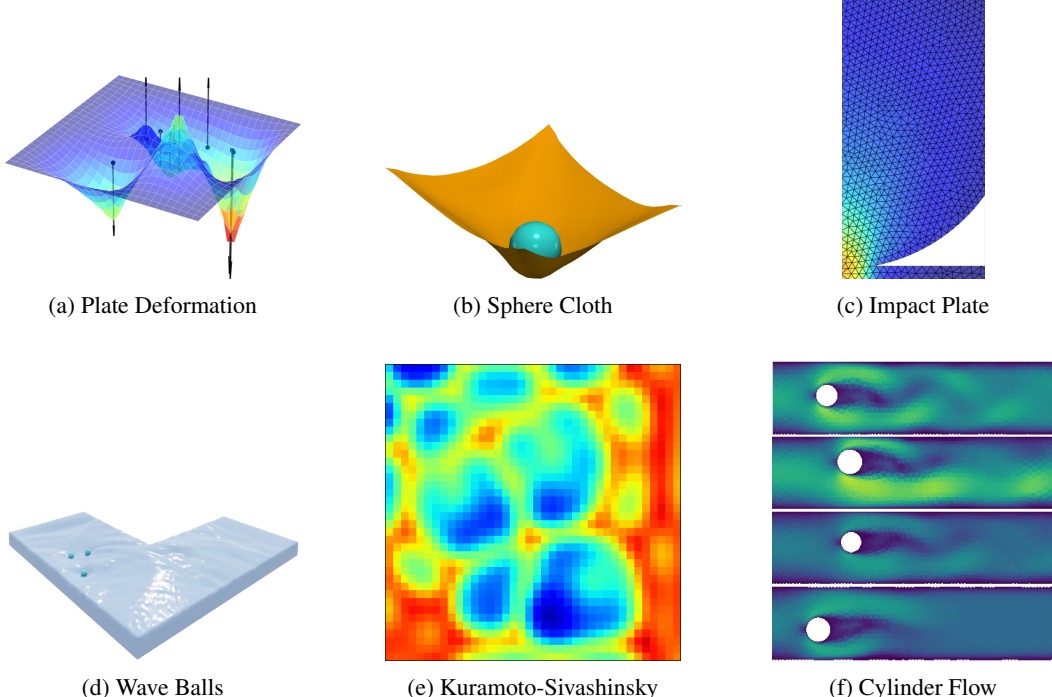

(a) Plate Deformation  (b) Sphere Cloth  (c) Impact Plate

(d) Wave Balls  (e) Kuramoto-Sivashinsky  (f) Cylinder Flow

Figure 4: Dataset overview. **(a) Plate Deformation**: a flat plate deformed subject to varying numbers and magnitudes of point forces, simulated with linear elasticity. **(b) Sphere Cloth**: a cloth mesh impacted by a falling sphere, producing elastic deformation with contact dynamics. **(c) Impact Plate**: an elastic plate under impact with a rigid ball. **(d) Wave Balls**: surface waves on water generated by three balls moving linearly from different initial positions. **(e) Kuramoto-Sivashinsky**: chaotic spatio-temporal evolution of a scalar field governed by the KS equation. **(f) Cylinder Flow**: incompressible fluid flow around cylindrical obstacles with vortex shedding.

## F  DATASETS DESCRIPTIONS

Table 4: Dataset summary: average number of nodes, average number of edges, total horizon $H$, window length $T$ (prediction sub-horizon), and train/val/test split (number of trajectories).

| Dataset | Avg. Nodes | Avg. Edges | Horizon $H$ | Window $T$ | Train/Val/Test |
|---|---|---|---|---|---|
| Plate Deformation | 851 | 3279 | 48 | 48 | 800/100/100 |
| Sphere Cloth | 401 | 3020 | 49 | 49 | 800/100/100 |
| Impact Plate | 2201 | 13031 | 51 | 51 | 2000/200/200 |
| Wave Balls | 1596 | 6048 | 50 | 50 | 250/100/50 |
| Cylinder Flow | 1885 | 10843 | 180 | 30 | 200/100/100 |
| Kuramoto-Sivashinsky | 1600 | 6240 | 300 | 100 | 250/100/50 |

Table 4 reports the dataset details, including the average number of nodes/edges, as well as horizon length and window size. For the full-rollout *Plate Deformation* task variant used in the ablation in Section 6, we predict the whole trajectory with length $T = 48$. In addition, for *Cylinder Flow* and *Kuramoto-Sivashinsky*, we follow the suggestion from Lienen & Günnemann (2022) to only split the sub-trajectories into equal-size, non-overlapping windows. Compared to the *Cylinder Flow* used in the Lienen & Günnemann (2022) paper, we use the window size of 30 instead of 10 to further increase the complexity.

Next, we report the node-level inputs and outputs used for each task in Table 5. For Lagrangian systems (Plate Deformation, Sphere Cloth, Impact Plate), the state consists of displacement $q_i$ and velocity $\dot{q}_i$, together with static properties $n_i$ such as material parameters, forces magnitude (for the

Table 5: Task Inputs/Outputs used in our experiments. $\boldsymbol{q}_i$: displacement, $\dot{\boldsymbol{q}}_i$: velocity, $\boldsymbol{n}_i$: static properties (e.g., node-type), $\rho_i$: density/scalar field, $\boldsymbol{v}_i$: velocity field (for Cylinder Flow). For methods training with multi-step loss ([MS]), they also receive initial absolute positions $\boldsymbol{x}_i^0$ (notated inline).

| Tasks | Type | Node Inputs | Node Outputs |
|---|---|---|---|
| Plate Deformation | Lagrangian | $\boldsymbol{n}_i$ | $\boldsymbol{q}_i$ |
| Sphere Cloth | Lagrangian | $\boldsymbol{n}_i,\ \dot{\boldsymbol{q}}_i,\ (\boldsymbol{q}_i^{(0)}$ [MS] $)$ | $\boldsymbol{q}_i,\ \dot{\boldsymbol{q}}_i$ |
| Impact Plate | Lagrangian | $\boldsymbol{n}_i,\ \dot{\boldsymbol{q}}_i,\ (\boldsymbol{q}_i^{(0)}$ [MS] $)$ | $\boldsymbol{q}_i,\ \dot{\boldsymbol{q}}_i$ |
| Wave Balls | Eulerian | $\boldsymbol{n}_i,\ \rho_i$ | $\rho_i,\ \dot{\rho}_i$ |
| Cylinder Flow | Eulerian | $\boldsymbol{n}_i,\ \boldsymbol{v}_i,\ \dot{\boldsymbol{v}}_i$ | $\boldsymbol{v}_i,\ \dot{\boldsymbol{v}}_i$ |
| Kuramoto-Sivashinsky | Eulerian | $\boldsymbol{n}_i,\ \rho_i$ | $\rho_i,\ \dot{\rho}_i$ |

Plate Deformation task) and node types. For these tasks, methods trained with the multi-step objective additionally receive the initial absolute position $\boldsymbol{q}_i^0$, which represents the rest configuration. For Eulerian systems (Wave Balls, Kuramoto-Sivashinsky), the state is represented by a scalar density field $\rho_i$, and the outputs include both $\rho_i$ and its temporal derivative $\dot{\rho}_i$. In Cylinder Flow, the state instead contains velocity $\boldsymbol{v}_i$, with $\dot{\boldsymbol{v}}_i$ provided to capture the temporal evolution of the velocity field.

In the following, we give the full details for each dataset, with examples given in Fig. 4:

**Solid mechanics.** This category includes Plate Deformation, Sphere Cloth, and Impact Plate (c.f. Yu et al. (2024)).

- *Plate Deformation:* The task is to predict the final state (or full horizon) of an initially flat plate subjected to external forces of varying positions and magnitudes. These forces are encoded as special nodes connected locally to the sheet within a small radius, and their number ranges from 1 to 19. This setup primarily evaluates the model's ability to capture long-range spatial interactions. Ground-truth simulations are obtained using linear-elastic dynamics with shell elements in the commercial Abaqus software (Smith, 2009).

- *Sphere Cloth:* A ball is dropped from random positions and heights onto a cloth mesh of size $19 \times 19$, and the system is simulated for $T = 50$ steps. To ensure fair training and compatibility with dissipative models, we further enhance the cloth mesh connectivity by introducing edges between the ball and every fourth cloth node. The dynamics are governed by elasticity, and the initial ball position strongly influences the outcome. Simulations are performed using multi-body physics in NVIDIA Isaac Sim (Mittal et al., 2023). We also consider a more challenging variant of this task, used to ablate the importance of the time-varying component, with a larger cloth size of $29 \times 29$ and rollout length $T = 100$.

  Note that this dataset has been redesigned from a similar scene introduced in Dahlinger et al. (2025), which was originally developed for a meta-learning setup, to instead probe long-range dependencies under direct supervised training.

- *Impact Plate:* This is a standard benchmark from the HCMT paper (Yu et al., 2024) to test long-range interaction, where the model must predict both stress and displacement propagation from distant nodes. The ground-truth simulation is Ansys.

**Fluid dynamics.** This category includes Wave Balls, Cylinder Flow, and the Kuramoto-Sivashinsky (KS) equation.

- *Cylinder Flow:* This task is based on the standard MeshGraphNets task (Pfaff et al., 2021), but we use only $H = 180$ time steps (vs. 600) and 200 trajectories (vs. 1000), windowed into non-overlapping segments of length 30. After 180 steps, the system converges to a periodic steady state. This setup highlights error accumulation under autoregressive training, explaining the lower MGN performance in our results.

- *Wave Balls:* We let three balls, starting with random positions, move from left to right linearly along a water surface in various grid shapes (cross, L, U, T). The dynamics are governed by a second-order hyperbolic PDE with external forcing generated by those balls:

$$\partial_{tt} u - c^2 \nabla^2 u = f_{\text{external}}(x, t),$$

The simulation is implemented in NVIDIA Warp (Macklin, 2022). An extended version of this task, used in the ablation study, runs for $T = 100$ steps. For the first 50 steps the balls move across the surface, and for the next 50 steps they stop. The waves, however, keep oscillating since there is no damping. This creates an interesting case where the model must learn not only the forced response but also continue the free oscillation, and figure out which phase the system is in.

- *Kuramoto-Sivashinsky (KS):* A 4th-order PDE commonly used in neural operator benchmarks (Li et al., 2021), generating chaotic behavior. Instead of random initial states, we use three Gaussian sources on a $40 \times 40$ grid and apply Neumann boundary conditions (not periodic) to focus on local rather than global behaviors. Simulated with the `py-pde` library (Zwicker, 2020). The KS equation in 2D is given by

$$\partial_t u = -\frac{1}{2}|\nabla u|^2 - \nabla^2 u - \nabla^4 u.$$

## G  HYPERPARAMETERS AND TRAINING TIME

Table 6: Hyperparameters used in all experiments. "AR" is autoregressive methods (e.g., MGN). "MS" is multi-step methods (others).

| Hyperparameter | Common | AR | MS |
|---|---|---|---|
| Node feature dimension | 128 | – | – |
| Latent representation dimension | 128 | – | – |
| Decoder hidden dimension | 128 | – | – |
| Optimizer | Adam | – | – |
| Learning rate | $5 \times 10^{-4}$ | – | – |
| Message-passing blocks | – | $15^{\dagger}$ | Window size |
| Training noise (std) | – | $1 \times 10^{-2}$ | none[§] |
| **Task-specific overrides (MS unless noted):** | | | |
| Plate Deformation | message blocks = 48. | | |
| Sphere Cloth | Warmup = 20. | | |
| Impact Plate | Warmup = 15. | | |
| Wave Balls | Warmup = 0. | | |
| Kuramoto–Sivashinsky | Warmup = 10; MS training noise = $1 \times 10^{-2}$. | | |
| Cylinder Flow | Warmup = 10; MS training noise = $1 \times 10^{-2}$. | | |

[†] MGN uses 15 message-passing blocks by default; Plate Deformation uses 50.
[§] For MS methods we disable training noise except on Cylinder Flow and Kuramoto–Sivashinsky, where we use $1 \times 10^{-2}$ due to varying initial conditions.

Table 6 summarizes the shared and task-specific hyperparameters. Most settings are common across models, with feature sizes fixed at 128 and Adam as the optimizer. For *MGN*, we adopt mean aggregation with Leaky ReLU, use 15 message-passing blocks by default (50 for Plate Deformation), and apply Gaussian training noise with $\sigma = 1 \times 10^{-2}$. In contrast, *MS* methods use ReLU activations in the encoder and decoder, while the graph-convolution blocks rely on $\tanh$ in A-DGN, IGNS and IGNS$_{\text{ti}}$. Training noise is generally disabled for *MS* methods, but we enable it with $\sigma = 1 \times 10^{-2}$ in Cylinder Flow and Kuramoto–Sivashinsky, where the initial conditions vary. In all cases, we normalize inputs and unnormalize outputs for every training batch, which stabilizes optimization and allows us to use the same learning rate and training noise across tasks.

Table 7 (a) shows the training time allocated to each task. To ensure fairness, all methods were trained for the same wall-clock time on a single NVIDIA A100 GPU. For the default settings, the budgets were chosen based on validation curves, where extending training further did not yield noticeable improvements. For the extended version of Sphere Cloth and Wave Balls tasks, we instead fixed a hard budget to ensure sufficient training for fair comparison in the ablation studies, independent of validation plateaus. In Table 7 (b), we report the number of parameters for each method. It is clear that the standard MGN with a non-shared processor requires by far the largest number of parameters.

Table 7: (a) Training time budget for each task, where all methods were trained with the same wall-clock time. (b) Parameter counts (measured on the Kuramoto-Sivashinsky task).

| Task | Default (hours) | Long (hours) |
|---|---|---|
| Plate Deformation | 12 | – |
| Sphere Cloth | 12 | 24 |
| Impact Plate | 48 | – |
| Wave Balls | 16 | 24 / 48 |
| Cylinder Flow | 24 | – |
| Kuramoto–Sivashinsky | 24 | – |

(a) Training time

| Method | #Parameters |
|---|---|
| **IGNS** | 216,000 |
| **IGNS$_{ti}$** | 80,100 |
| GCN+LN | 79,700 |
| MP-ODE | 101,000 |
| GraphCON | 67,800 |
| A-DGN | 100,000 |
| MGN (15 MP) | 1,800,000 |

(b) Parameter counts

| Method | Runtime (s) |
|---|---|
| MGN | $0.6690 \pm 0.0845$ |
| GCN_LN | $0.0381 \pm 0.0007$ |
| MP-ODE `RK4` | $0.1414 \pm 0.0068$ |
| MP-ODE `dopri5` | $0.3062 \pm 0.0064$ |
| ADGN | $0.0721 \pm 0.0008$ |
| GraphCON | $0.0433 \pm 0.0005$ |
| IGNS$_{ti}$ | $0.1253 \pm 0.0011$ |
| IGNS | $0.1703 \pm 0.0205$ |

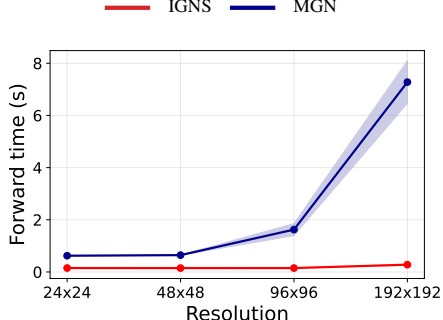

Figure 5: Runtime per forward pass on *WaveBalls* ($T$=200, batch size 1). Mean $\pm$ std over $4$ samples (seconds) on different methods **(left)** and with varying resolutions **(right)**.

## H  COMPLEXITY AND RUNTIMES

We discuss the theoretical complexity of our method, followed by a comparison of runtimes with other methods.

IGNS preserves the complexity of its underlying backbone GNN. For instance, when instantiated with a GCN backbone, each iteration is linear in the number of nodes $|V|$ and edges $|E|$, achieve a time complexity of $\mathcal{O}(|V| + |E|)$. Considering $L$ warmup steps and $T$ IGNS iterations, the total time computational cost is therefore $\mathcal{O}((L + T)(|V| + |E|))$. For comparison, MGN performs $S$ message-passing updates *per* simulation step, yielding $\mathcal{O}(S\,T\,(|V|+|E|))$; in our setup $S$=15, which explains why it is the slowest. GraphCON is simpler: it applies a single GCN only in the external-forcing branch, so its per-step cost remains $\mathcal{O}(|V|+|E|)$ with a small constant. By contrast, IGNS/IGNS$_{ti}$ uses three GCNs for $(\boldsymbol{q}, \boldsymbol{p})$ and forcing, raising the constant factor but increasing expressiveness, while still staying linear-time and *much faster than MGN*.

**Runtimes.** Fig. 5 reports the inference runtimes for one full-trajectory forward pass (in seconds) on *Wave Balls* with $T$=200 (batch size 1), measured on a single NVIDIA RTX 4080 (16GB). On the left, IGNS is competitive with other GNN backbones and substantially faster than adaptive ODE solvers and MGN; the latter is slowest because it performs inner spatial updates at every time step. On the right, we compare IGNS and MGN as we increase the Wave Balls resolution from $24\times24$ up to $192\times192$. This corresponds to graphs with roughly $450$ nodes at $24\times24$, about 6k nodes at $96\times96$, and around 23k nodes and 93k edges at $192\times192$. While the runtime of IGNS remains almost constant up to $96\times96$ and grows only moderately at $192\times192$, the runtime of MGN increases sharply and becomes an order of magnitude slower at the highest resolution. This behavior is consistent with the architectural design: MGN's additional spatial update, which is applied sequentially at every forward step, makes its cost scale with both the time horizon and the graph size, and prevents efficient GPU utilization compared to IGNS, which avoids these extra per-step spatial updates.

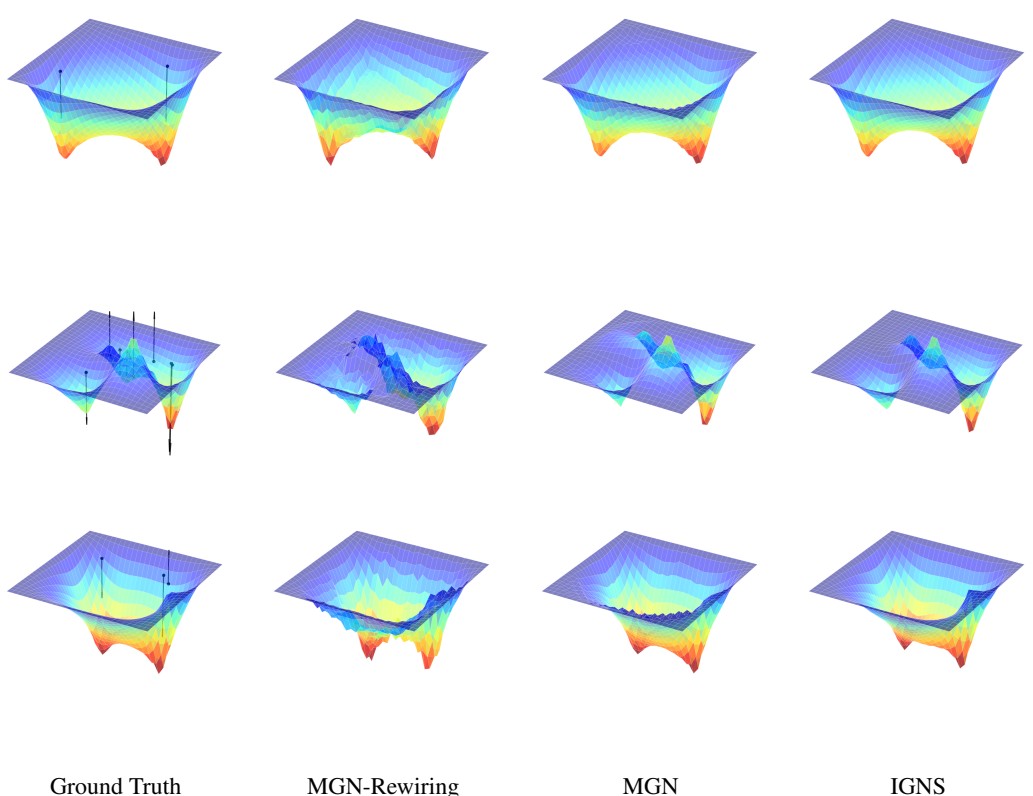

| Ground Truth | MGN-Rewiring | MGN | IGNS |

Figure 6: Plate Deformation qualitative comparison on three test samples with predictions from *MGN-rewiring*, MGN and IGNS. The arrows indicate the forces (with magnitude proportional to its length) applied to the plate at specific positions.

# I   ADDITIONAL RESULTS

Table 8: Comparison of train and test losses, shown as mean $\pm$ std, on Plate Deformation. Best results are highlighted with orange.

| Method | Test loss | Train loss |
|---|---|---|
| MGN ($m$=50) | $1.27 \pm 0.06$ | $0.09 \pm 0.01$ |
| MGN-rewiring ($m$=5) | $3.72 \pm 0.15$ | $0.20 \pm 0.01$ |
| **IGNS** | $1.34 \pm 0.05$ | $0.71 \pm 0.05$ |

**Plate Deformation.**   Fig. 6 shows qualitative results on three test samples, comparing IGNS, MGN, and a rewired variant of MGN. In MGN-rewiring, we improve connectivity by directly linking the force nodes to the entire mesh (through every second node), a common technique to mitigate over-squashing. We also reduce the number of message passing steps from 50 (standard MGN) to 5 (MGN-rewiring). The results highlight clear differences: only IGNS produces smooth and physically consistent surfaces, while MGN predictions are less smooth despite lower error, and MGN-rewiring generates the roughest outputs. Quantitative results in Table 8 further confirm this trend, showing that both MGN variants severely overfit by relying too heavily on geometric features, while IGNS achieves a better trade-off between train and test performance.

**Impact Plate.**   Table 9 reports quantitative results on the Impact Plate task, comparing IGNS and IGNS against the HCMT model from Yu et al. (2024) and MGN. Here, we take the results from Yu et al. (2024) for both HCMT and MGN, and report with the same metrics RMSE for position

Table 9: Impact Plate quantitative results. Reported are root mean squared errors (RMSE) for position and stress, shown as mean $\pm$ std. Best results are highlighted with orange.

| Method | Position RMSE | Stress RMSE ($\times 10^3$) |
|---|---|---|
| HCMT | $20.71 \pm 0.57$ | $14.74 \pm 5.02$ |
| MGN | $40.73 \pm 2.94$ | $35.87 \pm 11.89$ |
| **IGNS$_{\text{ti}}$** | $7.99 \pm 1.03$ | $3.25 \pm 0.69$ |
| **IGNS** | $7.65 \pm 0.94$ | $2.85 \pm 0.55$ |

Figure 7: Validation MSE on *SphereCloth-direct*. (a) IGNS core components: geometric encoding, warmup (multi-step loss is omitted as its curve explodes beyond the limit). (b) Damping and forcing in IGNS. (c) Damping and forcing in IGNS$_{\text{ti}}$.

and stress. Both IGNS and IGNS$_{\text{ti}}$ significantly outperform the baselines in both position and stress prediction, with IGNS achieving the best performance overall. This demonstrates the effectiveness of our approach in capturing long-range interactions and complex dynamics in this challenging task.

Table 10: Sphere Cloth quantitative results. Reported are mean squared errors (MSE) for overall loss, position, and velocity, shown as mean $\pm$ std ($\times 10^{-3}$). Best results are highlighted with orange.

| Model | Loss | Position | Velocity |
|---|---|---|---|
| GCN+LN | $26.45 \pm 0.23$ | $4.14 \pm 0.04$ | $22.31 \pm 0.20$ |
| MGN | $32.07 \pm 2.45$ | $6.53 \pm 0.76$ | $25.54 \pm 1.73$ |
| MP-ODE | $25.75 \pm 1.32$ | $4.26 \pm 0.26$ | $21.49 \pm 1.06$ |
| ADGN | $30.99 \pm 0.80$ | $4.57 \pm 0.25$ | $26.42 \pm 0.66$ |
| GraphCON | $29.00 \pm 0.44$ | $4.19 \pm 0.07$ | $24.82 \pm 0.40$ |
| **IGNS$_{\text{ti}}$** | $28.20 \pm 0.35$ | $3.74 \pm 0.08$ | $24.46 \pm 0.30$ |
| **IGNS** | $24.16 \pm 0.67$ | $3.55 \pm 0.15$ | $20.60 \pm 0.55$ |

**Sphere Cloth.** In this task, the model must jointly predict the motion of a rigid sphere (represented only by its center position) and a deformable cloth, creating strong coupling dependencies. Figure 9 illustrates two representative test samples. In the top sample, MGN produces wrong predictions due to error accumulation, while in the bottom sample it manages to produce a more reasonable result. This highlights the instability of first order methods, in contrast to IGNS, which consistently yields results closest to the ground truth. The quantitative results in Table 10 confirm this trend: although IGNS$_{\text{ti}}$ achieves slightly lower position error, its surfaces remain rougher, while MGN is prone to error accumulation and overfitting. Other baselines (GraphCON, ADGN, and GCN) fail to generate smooth surfaces.

**Sphere Cloth direct.** In Fig. 7, the validation curves show that even without forcing, the time-varying model (IGNS) can reach a reasonable solution after a longer training phase, which explains the larger variance across seeds. This suggests that time variation can partly substitute for external

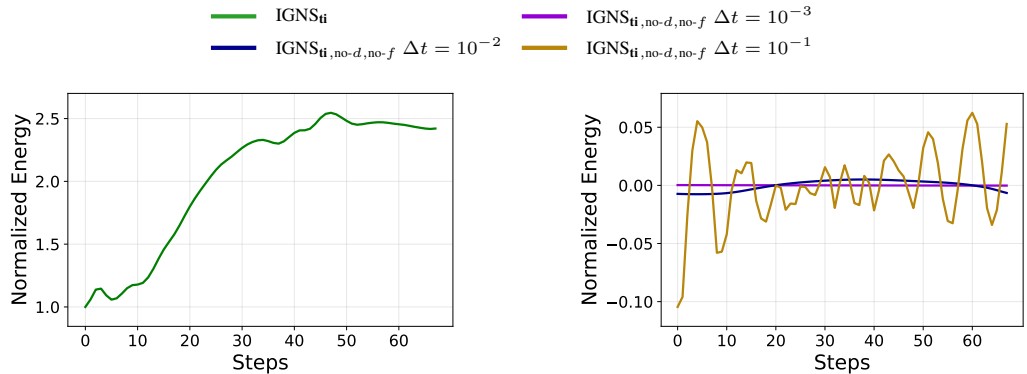

Figure 8: Energy behavior on *SphereCloth-direct*. **Left:** IGNS$_{\text{ti}}$ (with damping and forcing) shows an increasing energy profile, as expected for a non-conservative model that learns external actuation/dissipation. **Right:** removing damping and forcing yields an approximately conservative evolution; smaller time steps ($\Delta t \in \{10^{-3}, 10^{-2}\}$) keep energy nearly constant, while a larger step ($\Delta t = 10^{-1}$) produces mild oscillations due to discretization error.

forcing by mediating energy exchange and therefore can help increasing the model's expressiveness. By contrast, the time-invariant model (IGNS$_{\text{ti}}$) depends heavily on forcing: when it is removed, optimization plateaus at a poor local optimum for a long training time (until $400$ epochs), and then starts decaying with a slower rate compared to the time-varying versions. These behaviors are consistent with the Test MSE reported in Table 3. **Energy conservation analysis.** Next, to assess energy behavior, we load the best validation checkpoint of IGNS$_{\text{ti}}$ on this task. In Fig. 8, the full port-Hamiltonian model with all the terms (left) increases normalized energy over time, reflecting learned non-conservative effects. When we disable forcing and damping (right) and vary the integrator step, the dynamics become nearly conservative: the energy stays flat for small $\Delta t$ and oscillates slightly for larger $\Delta t$. This aligns with our ablations, where forcing acts as the main driver of energy injection while the time-invariant core without non-conservative terms maintains near-constant energy.

**Wave Balls.** Fig. 10 compares IGNS with the baselines MGN, GraphCON, and A-DGN. The absolute error maps show that IGNS is the only method producing consistently reasonable results. MGN is able to track the ball positions but overestimates wave amplitude, leading to large errors. GraphCON captures the wave pattern but mislocalizes the balls, causing errors near the sources. A-DGN fails to learn this task.

**Wave Balls (long)** ($T{=}100$ and $T{=}200$). We generate trajectories with a high-resolution simulator using a ground-truth horizon $T_{\text{gt}}{=}400$ and form two settings by subsampling: $T{=}100$ ($4\times$) and $T{=}200$ ($2\times$). This probes whether models can learn under larger step sizes while remaining accurate over long horizons. As shown in Table 11, MGN fails severely on this task due to over-reliance on autoregressive state. In contrast, both IGNS and IGNS$_{\text{ti}}$ perform well; IGNS attains the lowest error and captures finer wave-density details than its time-invariant counterpart.

Table 11: Comparison of *test* losses on *WaveBall* at $T{\in}\{100, 200\}$ (mean $\pm$ std; lower is better). Values are reported in $\times 10^{-3}$. Best results are highlighted with **orange**. MGN is clearly worse, and IGNS surpasses the time-invariant variant IGNS$_{\text{ti}}$, confirming the benefit of the time-varying port-Hamiltonian core.

| Method | $T{=}100$ | $T{=}200$ |
|---|---|---|
| MGN | $1.644 \pm 0.096$ | $2.469 \pm 0.221$ |
| GraphCON | $2.476 \pm 0.082$ | $2.011 \pm 0.063$ |
| **IGNS$_{\text{ti}}$** | $0.675 \pm 0.014$ | $0.498 \pm 0.022$ |
| **IGNS** | $0.549 \pm 0.014$ | $0.402 \pm 0.037$ |

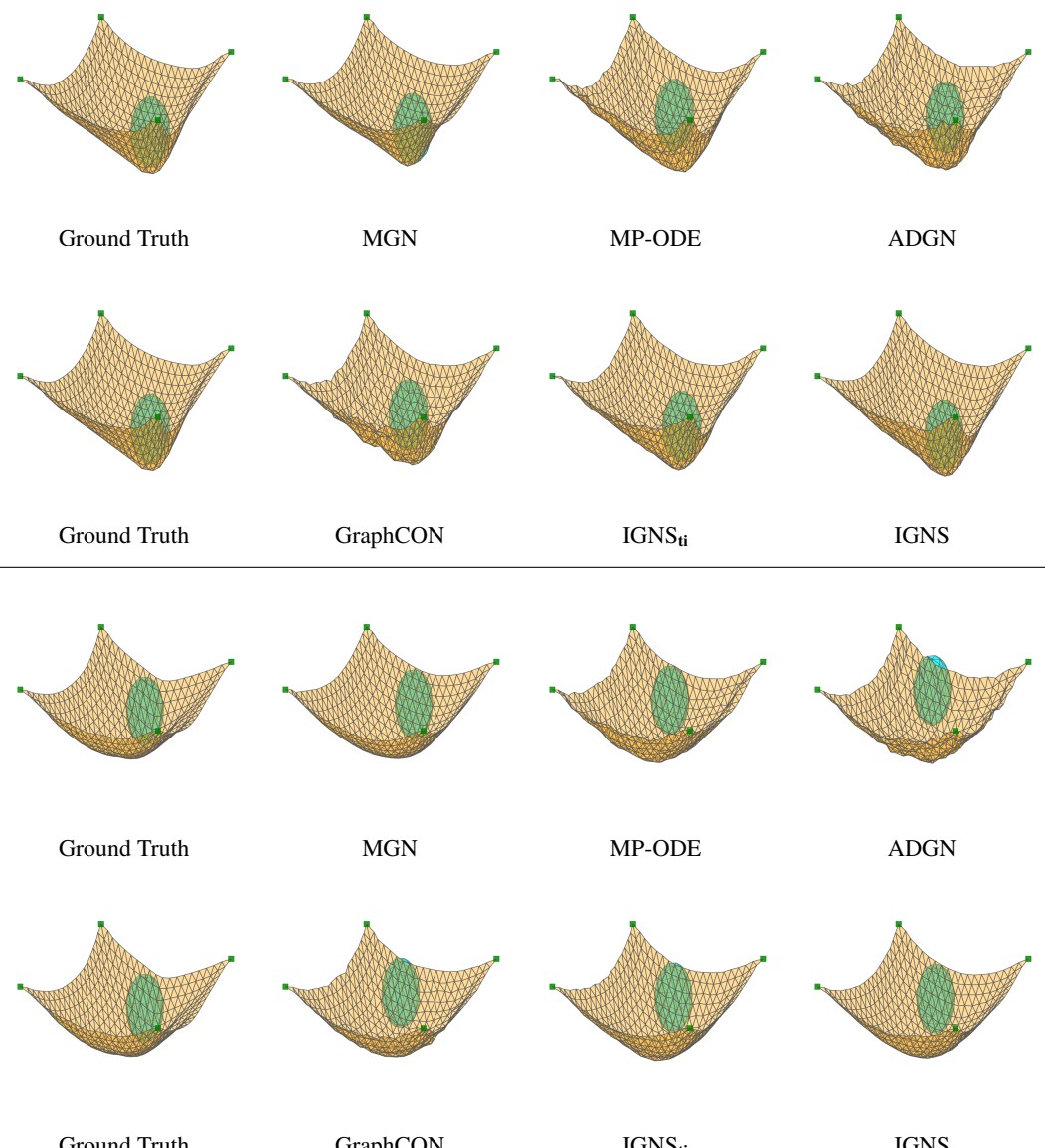

Figure 9: Sphere Cloth qualitative comparison on two test samples (separated by a horizontal line) with predictions at the final step $t = T$ from first-order models (MGN, MP-ODE) and second-order models (ADGN, GraphCON, **IGNS$_{ti}$** and **IGNS**).

**Kuramoto-Sivashinsky.** To illustrate performance, we show predictions and absolute errors for FNO-RNN, MGN, GraphCON, and IGNS on two test samples (Figs. 11 and 12). The first sample reflects an early stage where local behavior dominates; the second shows a later, more chaotic regime. Two patterns emerge: FNO-RNN produces more globally distributed responses, while the graph-based models remain more localized. In both samples, MGN shows larger absolute errors than the second-order models (GraphCON and IGNS). In Fig. 11, MGN also misses one mode, consistent with error accumulation under autoregressive rollout. Overall, GraphCON performs best on this task, with IGNS and IGNS$_{ti}$ close competitors.

**Warmup Phase.** Figure 13 illustrates the effect of different warmup steps $l$ on the Plate Deformation task over a full rollout. With almost no warmup ($l = 1$), the model produces sharp local patterns that gradually recover towards the correct global shape, matching the ground truth only after about

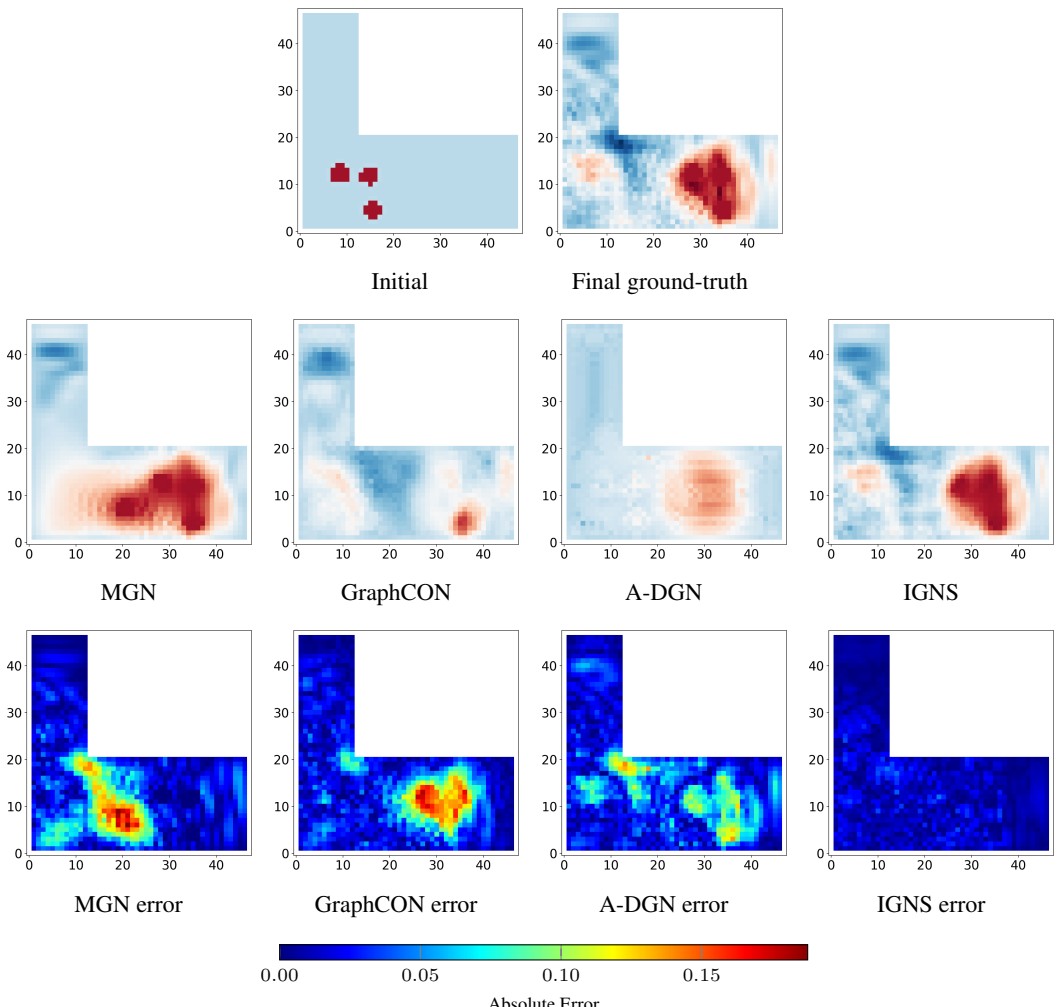

Figure 10: Wave Balls qualitative comparison on one test sample. Top: initial condition and ground-truth final state at time $t=T$. Middle: predictions at $t=T$ from MGN, GraphCON, A-DGN and **IGNS**. Bottom: absolute error maps, $|u_{\text{pred}} - u_{\text{gt}}|$. The horizontal colorbars indicate the error magnitude in the bottom row.

$t = 10$. Increasing $l$ accelerates this recovery: with $l = 5$, the deformation aligns well already by $t = 5$, and with $l = 10$ by $t = 3$. For larger values, $l = 30$ and $l = 50$, the global shape is already close to the ground truth at $t = 1$. These results highlight the role of the warmup phase in seeding long-range information before rollout, enabling the model to capture the dynamics more effectively from the very first step.

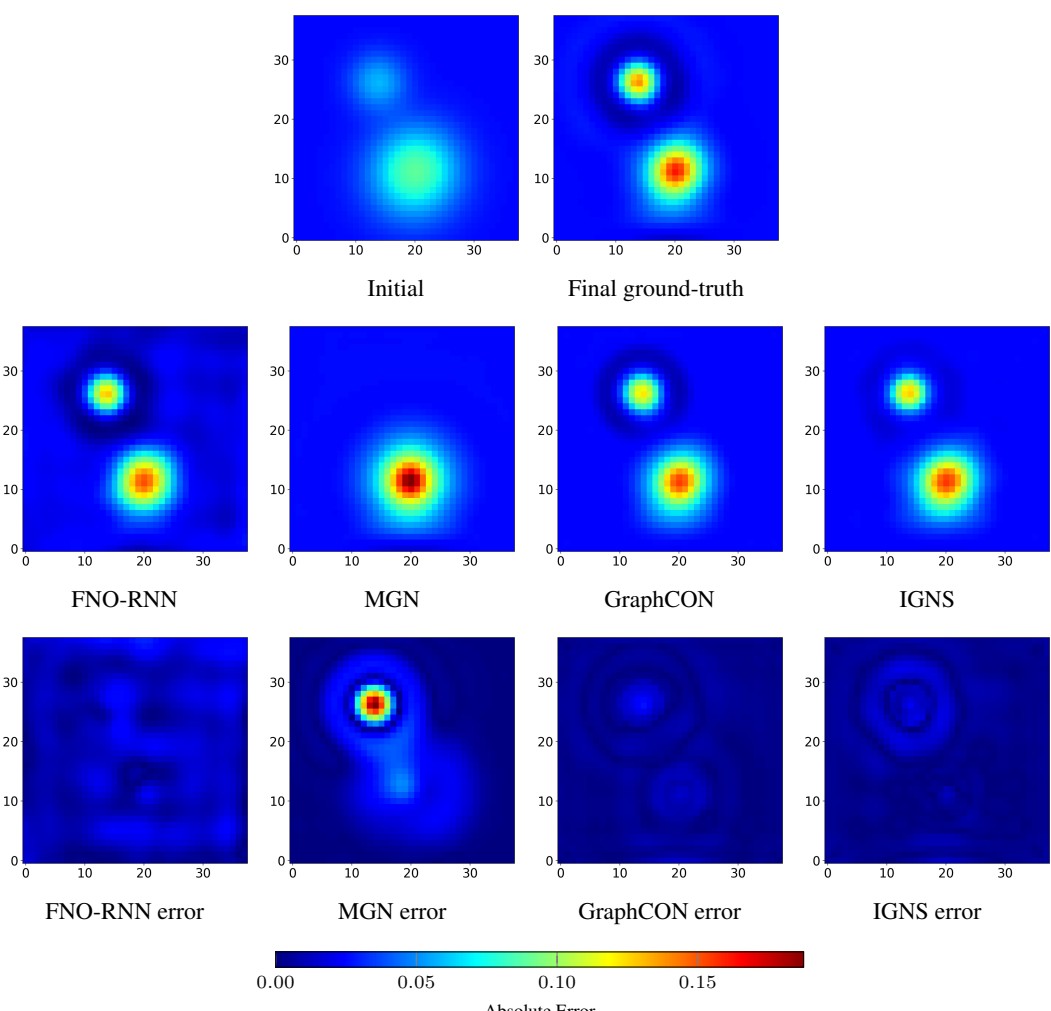

Figure 11: Kuramoto-Sivashinsky (KS) qualitative comparison on one test sample. Top: initial condition and ground-truth final state at time $t{=}T$. Middle: predictions at $t{=}T$ from FNO-RNN, MGN, GraphCON, and **IGNS**. Bottom: absolute error maps, $|u_{\text{pred}} - u_{\text{gt}}|$. The horizontal colorbars indicate the error magnitude in the bottom row.

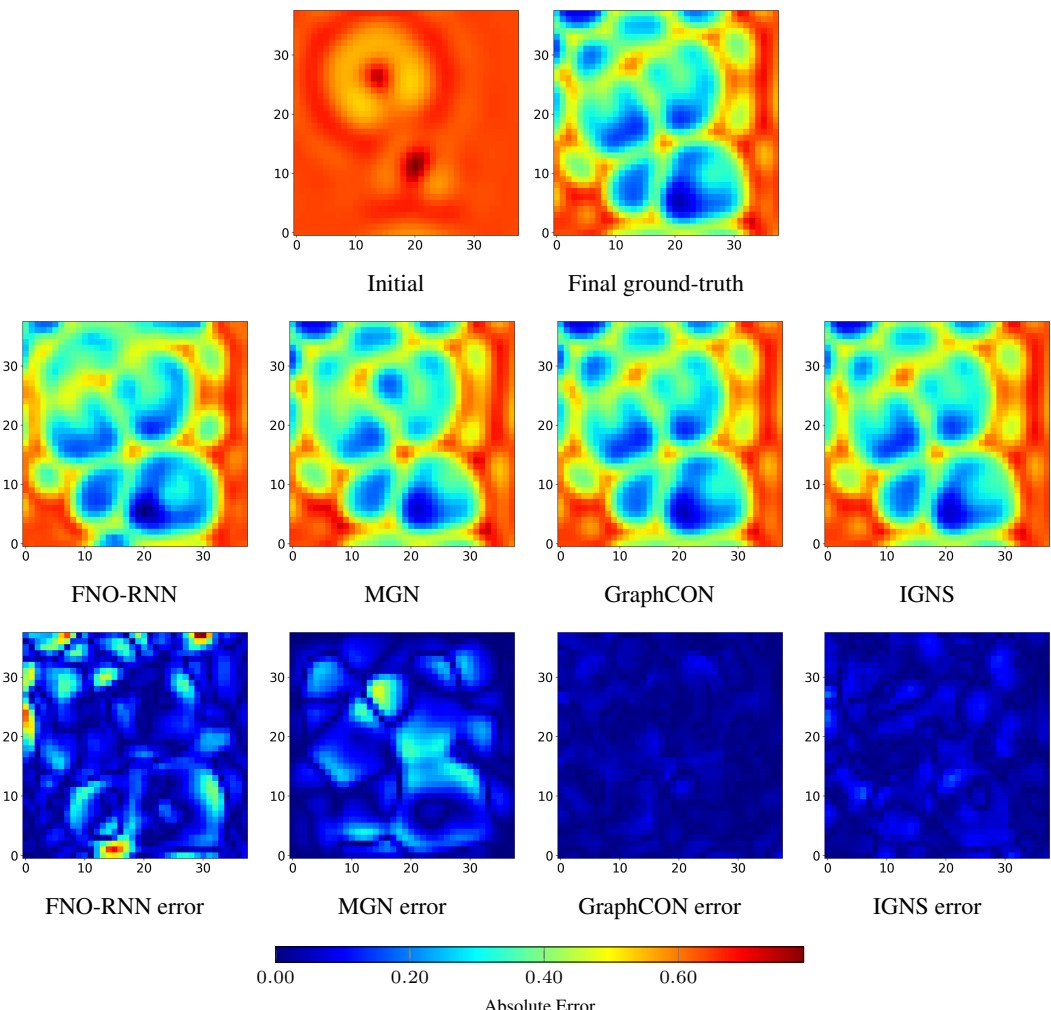

Figure 12: Kuramoto-Sivashinsky (KS) qualitative comparison on one test sample. Top: initial condition and ground-truth final state at time $t=T$. Middle: predictions at $t=T$ from FNO-RNN MGN, GraphCON, and **IGNS**. Bottom: absolute error maps, $|u_{\text{pred}} - u_{\text{gt}}|$. The horizontal colorbars indicate the error magnitude in the bottom row.

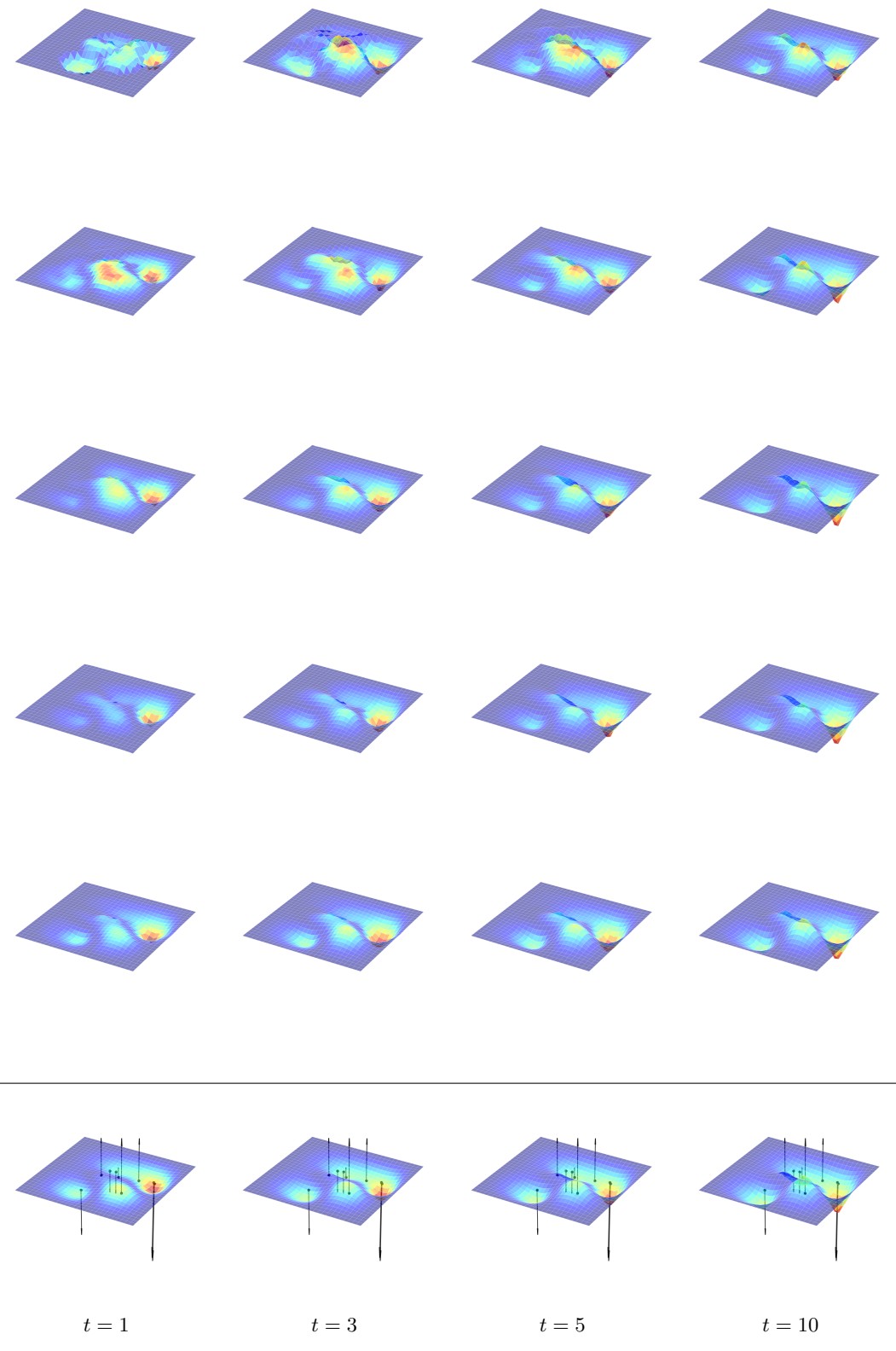

$t = 1$ $\quad\quad\quad\quad$ $t = 3$ $\quad\quad\quad\quad$ $t = 5$ $\quad\quad\quad\quad$ $t = 10$

Figure 13: Comparison between different warmup steps $l$ from top to bottom $l = \{1, 5, 10, 30, 50\}$ with ground truth in the last row, on Plate Deformation task.

