# OpenReview forum: "Improving Long-Range Interactions in Graph Neural Simulators via Hamiltonian Dynamics"
_ICLR.cc/2026/Conference — ICLR 2026 Poster_

### Official Review · Reviewer_S8R7 · 2025-10-19

**Soundness:** 3
**Presentation:** 3
**Contribution:** 3
**Rating:** 4
**Confidence:** 4

**Summary:**

This paper introduces Information-preserving Graph Neural Simulators (IGNS), a graph-based neural simulator that improves modeling of complex physical systems. IGNS enforces Hamiltonian dynamics to preserve long-range interactions and extends to non-conservative systems. It includes warmup initialization, geometric encoding, and multi-step training to enhance stability. Evaluated on new benchmarks with long-range dependencies and external forces, IGNS outperforms state-of-the-art methods in accuracy and robustness for dynamic systems.

**Strengths:**

- The proposed Information-preserving Graph Neural Simulator introduces a principled integration of port-Hamiltonian dynamics into graph-based simulators, marking a significant step beyond existing message-passing and oscillatory GNN frameworks.
- Theoretical analyses are thorough and provide a clear justification for the model’s ability to capture complex and long-range physical interactions.
- Experimental evaluation is comprehensive, spanning six datasets and consistently demonstrating the superior accuracy and stability of IGNS compared to strong baselines.
- The paper is clearly written and well structured: the motivation for each component (port-Hamiltonian core, warmup phase, geometric encoding, and multi-step loss) is clearly articulated, and the accompanying figures effectively convey both the methodology and the empirical findings.

**Weaknesses:**

- Although the theoretical analysis establishes information preservation and universality, it remains largely qualitative in linking these properties to the observed empirical improvements. A more quantitative or ablation-based verification (e.g., measuring gradient norms or energy conservation over rollouts) would provide stronger evidence for the theoretical claims.
- The training/testing computational overhead of the port-Hamiltonian formulation and the warmup phase is not explicitly analyzed; reporting runtime or memory costs relative to standard GNSs would clarify the practical trade-offs.
- Although the benchmarks are diverse, most tasks are synthetic or controlled simulations. It would strengthen the paper’s significance to include or discuss applications in more realistic or large-scale physical systems.
- The geometric encoding used to map edges to features follows the same formulation as previous works (e.g., MGN) and therefore cannot be considered a novel contribution.
- Several related approaches are not cited or compared [1–4], which limits the contextual positioning of this work within recent advances in graph-based physical simulation.
- Introducing "warmup phase" in GNNs is not novel. Eagle [2] employs a warmup-like phase in its encoder, using multiple message-passing blocks to aggregate local and global context before rollout, which parallels the proposed initialization strategy.
- The separation of state variables into coordinates and momenta, as well as the coordinate–momentum supervision in Eq. (10), are established techniques already used in [2–3].

[1] EvoMesh: Adaptive Physical Simulation with Hierarchical Graph Evolutions. ICML 2025

[2] Eagle: Large-Scale Learning of Turbulent Fluid Dynamics with Mesh Transformers. ICLR 2023

[3] Efficient Learning of Mesh-Based Physical Simulation with BSMS-GNN. ICML 2023

[4] Physics meets Topology: Physics-informed topological neural networks for learning rigid body dynamics

**Questions:**

1. In L201-203 and in Appendix D, the paper states that $\gamma_\theta(t)$ and $\tau_\theta(t)$ are time-varying coefficient vectors produced by MLPs with parameters θ. Could the authors clarify what exactly is used as the input t to these MLPs? Is t normalized to a fixed range (e.g., [0, 1]) or directly represented as the raw timestep index? Additionally, if the model is trained on trajectories with 400 steps, can the learned time-dependent MLPs generalize to longer rollouts (e.g., 1000 steps) without retraining, or does the model rely on an absolute temporal scale?
2. L231-233 states: "Thanks to the energy conserving core of IGNS, this globally informed latent state is preserved throughout the rollout, rather than being dissipated." Could the authors clarify why this happens? Can you provide qualitative or quantitative analysis showing how the latent state is preserved over time？
3. Regarding the multi-step loss: how many time steps were included in the loss computation? Are the reported results based on single-step MSE or on the rollout of the entire sequence? Please include these experiment details.
4. Why does WaveBall not require a warmup phase?
5.  In the supplementary code (`igns.py`), L651: `x = self.one_step(x, edge_index, edge_weight, batch, t=i)` passes `i` as `t`. Here, `i` corresponds to the layer index rather than the time step. Can the authors explain why this is done?

---

> ### Author Response · Authors · 2025-11-21
> **Response to Reviewer S8R7 - Part 1**
>
> We thank the reviewer for their time and effort and for acknowledging our IGNS model is a principled framework, the thorough theoretical analyzes, and the comprehensive experiments, as well as the clear presentation. Below, we provide answers to the reviewer's concerns.
>
> > A more quantitative or ablation-based verification…
> >
>
> We thank the reviewer for the suggestions for new empirical analyzes to verify the theoretical properties of IGNS. We now provide an ablation called **energy conservation analysis** in App. I. In Fig. 7, the right panel (no forcing and no damping) shows that the normalized energy oscillates over time and, for smaller step sizes, stays close to zero. In contrast, the left panel (full port-Hamiltonian with forcing and damping) allows energy exchange and therefore increases the energy.
>
> Fig. 6 further supports the role of non-conservative effects: on the same task, the time-varying IGNS without forcing or damping can still reach a good solution after longer training, whereas the time-invariant IGNS_ti shows a shallow learning curve due to its conservative behavior. This suggests that time-varying weights enable energy exchange and make the model more expressive.
>
> > The training/testing computational overhead…
> >
>
> We added a complexity analysis and a runtime table in the App. H. In general, IGNS yields a similar computational cost as its GNN backbone, e.g., with a GCN backbone, one forward step is $\mathcal{O}(|V|+|E|)$ time. When incorporating $L$ warmup steps and $T$ rollout steps, the total cost is $\mathcal{O}(L+T)(|V|+|E|)$ time. For comparison, MGN requires $S$ spatial-updates per time-integration step, yielding $\mathcal{O}(ST(|V|+|E|))$ and explaining its highest runtime among other evaluated methods on a long-horizon WaveBalls task ($T=200$), see Tab. 8.
>
> > more realistic or large-scale physical systems would strengthen the paper’s significance
> >
>
> We thank the reviewer for pointing this out. As we intend to test out the new Graph Neural Simulators, in the paper we have considered the Impact Plate, Cylinder Flow and Kuramoto-Sivashinsky as the standard benchmarks for this setting. We are happy to run more realistic tasks if the reviewer can point us to them.
>
> > The geometric encoding, warmup phase, separation of state variables into coordinates and momenta are not novel.
> >
>
> We thank the reviewer for these comments. We agree there are similarities to prior work in the geometric encoding, the warmup phase, and the split into coordinates and momenta. However, we did not claim these individual components as novel. Our aim in this paper is to explain their roles in our framework with theoretical and empirical evidence, further supported by new ablations (Tab. 3). We have revised the paper to clarify this concern and highlight the novelty. We summarize it below.
>
> Our primary goal remains to improve long-range dependencies in Graph Neural Simulators. To this end, we use a port-Hamiltonian system as the dynamics core. We show (Theorem 1) that integrating this system can approximate the target dynamics. This enables **PDE matching**, where we can **supervise the full trajectory with a fixed-window $T$ using multi-step loss at both training and testing**. This mirrors classical PDE solvers, where information propagates in space as time advances.
>
> Empirically, we show that our method is able to train on the original mesh/graph space over long window $T$ (WaveBall up to $T=200$). We recall that prior GNS work typically uses short horizons (e.g., [1] trains and tests with $T=10$) or trains with a short multi-step window and then rolls out autoregressively (e.g., [2] uses $T=8$), which leads to error accumulation [3] or applies multi-step supervision only in a reduced latent space rather than on the original graph [4]. This is the key difference we stress.
>
> The warmup phase and geometric encoding are then supporting mechanisms. More explicitly, warmup broadcasts context globally, and geometric encoding handles irregular meshes. In the revision, we clarify this scope and add Tab. 3 ablations to study the roles of these components.
>
> [1] Learning the Dynamics of Physical Systems from Sparse Observations with Finite Element Networks. ICLR 2022
>
> [2] Eagle: Large-Scale Learning of Turbulent Fluid Dynamics with Mesh Transformers. ICLR 2023
>
> [3] Message Passing Neural PDE Solvers. ICLR 2022
>
> [4] Predicting Physics in Mesh-reduced Space with Temporal Attention. ICLR 2022

---

> ### Author Response · Authors · 2025-11-21
> **Response to Reviewer S8R7 - Part 2**
>
> > Several related approaches are not cited or compared
> >
>
> We thank the reviewer and revise the paper with updated citations for the mentioned paper in the Related Work section in App. B. In general, these methods address long-range spatial dependencies via hierarchical structures and/or Transformers, which typically incur quadratic cost in the spatial size (e.g., $O(N^2)$ time/memory for attention, where $N$ depends on the chosen space). In contrast, IGNS keeps the backbone’s linear complexity and operates at node level on the original graph: one step is $O(∣V∣+∣E∣)$, and with $L$ warmup and $T$  rollout steps the total is $O((L+T)(∣V∣+∣E∣))$. This preserves scalability in both space and time while targeting the same long-range goal. In addition, they generally do autoregressive, which is prone to error accumulation.
>
> > Q1 and Q5: input $t$ in the MLPs for time-varying coefficients, and its generalization ability to longer rollouts
> >
>
> We input the raw step index $t$ and encode it with a standard spectral (sin-cos) positional embedding. This allows the model to accept times beyond the training window. However, in practice, long extrapolation degrades the performance due to the error accumulation. This is exactly the settings we consider in our paper, where autoregressive rollouts fail severely under our tasks, which exhibit complex dynamics such as Wave Balls and Kuramoto-Sivashinsky.
>
> > Q2: clarification on energy conserving core of IGNS
> >
>
> We studied the warmup phase extensively and quantified the effect in Fig. 3. Qualitative examples are in Fig. 12 (with GIFs in the supplement). A larger warmup yields better globally consistent states already at $t=1$. We additionally show the energy behavior of the pure Hamiltonian core in Fig. 7, where the conservative system keeps normalized energy nearly constant. Together, these results support the claim that the globally informed latent state is not dissipated through the rollout.
>
> For a more detailed answer: the Hamiltonian core limits numerical dissipation, so the latent does not “forget” information across steps. This is a useful property if the task needs global context at $t=1$ (common when observation frequency is lower than simulation frequency), one message-passing step is not enough because it only reaches one hop. Using $L$ warmup steps expands the receptive field to $L$ hops, so with a large enough $L$ the initial latent becomes globally informed. After that, the energy-conserving core maintains this information during the rollout.
>
> > Q3: multi-step loss
> >
>
> As our goal is to avoid error accumulation from autoregressive rollout, we fix the prediction window for both training and evaluation. All reported MSEs are **full-window** losses over that fixed window (not single-step). Task-specific horizons/windows are listed in Table 4.
>
> > Q4: Why does WaveBall not require a warmup phase?
> >
>
> Warmup helps when the task needs global information at $t=1$. For WaveBalls, we intentionally design the setup to not require immediate global aggregation. The goal is to probe oscillatory behavior under external forcing.

---

> ### Author Response · Authors · 2025-11-27
> **Follow-up on Review Feedback**
>
> Thank you again for your thoughtful review. We appreciate your recognition of the principled nature of our method, the thorough theoretical analysis, the comprehensive experimental evaluation, and the overall clarity of the paper. We’re following up since we haven’t seen a response to our rebuttal yet, and we’d be happy to discuss further before the discussion phase ends.
>
> To briefly summarize, we provided:
>
> - An energy conservation analysis (App. I, Fig. 7) showing conservative vs. full port-Hamiltonian behavior, plus a validation-curve (Fig. 6).
> - A complexity analysis + runtime table: IGNS shows that it is faster than MGN.
> - A clarification of novelty: Our core contribution, which we believe is novel, is the PDE-matching setup. Here, by solving the port-Hamiltonian dynamics and supervising it with the ground-truth trajectories, we can improve the long-range interactions of GNS across space and time, thereby reducing error accumulation stemming from one-step losses and autoregressive rollouts.
> - A geometric-encoding ablation: with vs. without static geometric encoding on *SphereCloth-direct* (Fig. 6a, Tab. 3a). The model can converge without it but is more accurate with it. This helps clarify its role in our full model IGNS.
> - A discussion on the related work the Reviewer has mentioned.
> - The clarifications and ablations addressing your questions (time input/encoding, warmup, full-window multi-step loss).
>
> We hope these address your concerns. We’re happy to follow your guidance on any remaining points and, given the added analyses and clarifications, kindly ask you to consider revisiting your score.

---

### Official Review · Reviewer_eSDm · 2025-10-27

**Soundness:** 2
**Presentation:** 2
**Contribution:** 2
**Rating:** 2
**Confidence:** 4

**Summary:**

The paper introduces a graph neural simulator built on a port-Hamiltonian system. Unlike the existing relevant literature, the core idea of the proposed method is to introduce the symplectic integrator while proposing a port Hamiltonian involving non-conservative energy terms in order to closely align the proposed emulator with the ODE dynamics. The warmup iteration is also proposed in an attempt to enhance the long-range message propagation. The paper also conducts theoretical analysis about the universality and the sensitivity of the model. The proposed framework is evaluated on a range experiment including a couple of new scenarios designed to assess the long-range propagation capability under external forcing.


Overall, I think the paper is not ready for the publication in the current form, because 1) the main arguments about the theoretical analysis and proposed architecture are over-claimed and loose, and 2) the experiments miss some relevant baselines and essential ablation study. In particular, Theorem 2 is almost completely identical to a theoretical result in an existing literature. A detailed clarification is necessary to highlight the difference. The detail of the warmup iteration is also missing, which is another factor that causes the difficulty to assess the significance and soundness of the contribution of the paper. The details are given in "Weaknesses" and "Questions" columns.

**Strengths:**

- The universality result (although I do not fully understand the proof yet) is novel.
- Proposal of new tasks, specifically designed to test long-range propagation and oscillatory dynamics under external forcing.

**Weaknesses:**

**Over-claimed assertions:**
- A sentence starting at line 290 regarding Theorem 1 is over-claiming. Being with/without compact supports makes a huge difference in the significance.
- The multi-step objective is a pretty common loss function for training auto-regressive models. The authors need to cite relevant papers or argue that this is a pretty common approach. The use of symplectic integrator in this context is also not novel.
- Theorem 2 is almost identical to a sensitivity result in [1], and it is unclear if this theorem is different enough from the result of [1] to claim it as novel and/or original.

**Misdirected experiments:**
- The core idea of the paper is closely related to the idea in [1], but apparently the baselines in the main experiments do not include the model, which makes the experiments setting look unfair and the result unfairly unconvincing.
- The ablation version of damping and forcing/residual terms is missing. This ablated model is essentially equivalent to the idea in [1] adopted to symplectic integrator with well-adopted multi-step loss, which I believe is valuable to be compared.
- The other fundamental difference is use of symplectic integrator and warmup iteration, but the ablation experiment also misses these aspects.
- The paper addresses the long-range propagation problem, but the experiment misses the impact of increasing the resolution of the space, which controls the difficulty of the long-range message propagation.

**Inaccurate description on the assertion and proof of Theorem 1:**
- Line 287: $\dot{x}_{0}$ should not be included.
- Theorem 1 should need a compact support on which $F$ is approximated by $\Psi_{\theta}$.
- Line 855: The second Hamiltonian equation. This should be introduced formally.
- Line 1028 misses the definition of $\bar{p}$ and the relation between $p$ and $q$ is unclear, so the derivation of (24) is non-trivial.

**Minor:**
- Typo: the map $\Phi$ is duplicated in line 987
- Diameter is important metric in this context, but it is missing from Table 3


[1] Heilig, et.al., "Port Hamiltonian Architectural Bias for Long-Range Propagation in Deep Graph Networks.", ICLR 2025.

**Questions:**

**Regarding the warmup iteration:**
- How exactly does the warmup iteration work? I cannot find its detail including the update formula for each iteration.
- Is the port-Hamiltonian used in the warmup phase shared with the time-evolving forward model?

**Proof of Theorem 1:**
- Why is it fine to set $D$ to be $0$ without the loss of generality? It is not obvious at first glance.
- What does B represent at line 989. Is it a parameter involved in $r(t)$?

---

> ### Author Response · Authors · 2025-11-21
> **Response to Reviewer eSDm - Part 1**
>
> We thank the reviewer for their time and effort and for acknowledging the novelty in the universality result, and the proposal of new tasks. Below, we provide answers to the reviewer's concerns.
>
> > Theorem 1 is over-claiming
> >
>
> We thank the reviewer for this remark. We are aware of this assumption, which is part of the Universality Theorem for Neural Oscillators (Theorem 3 of our manuscript, Theorem 3.1 of [1]), which we included in that statement. For clarity and precision, we specify this assumption in the statement of Theorem 1 and in the following discussion as well. Compactness is required in every Universality theorem, both for neural networks (that is, for learning maps between $\mathbb{R}^p$ and $\mathbb{R}^d$) and operator-like learning (the case of Neural Oscillators, where we learn a map between functions).
>
> For a short answer, this assumption is not particularly restrictive for our purposes, that is, mainly physical simulations, as it is enough to consider physical quantities with limited values (finite scale) and an upper bound on energy (feasible in real systems). Both these assumptions are fair for physical systems.
>
> For a more detailed answer, the assumption is to have a compact subset K of the space $\mathbb{R}^{n\times d})$ (that is, the set of initial conditions). For this, it is sufficient to have K closed and limited, that is, there exists $A>0$  such that  $|q_0|,|p_0|\leq A$. For a physical interpretation, this means that the scale of the physical quantities is limited and that there is an upper bound on the energy in our simulations. We think that both these assumptions are fair for contained physical simulations.
>
> We included this discussion in the revised version of the paper, and we think this shows that our Universality Theorem is very applicable to our settings.
>
> > The multi-step objective is a pretty common loss…
> >
>
> We thank the reviewer. To clarify, we did not claim the multi-step objective as novel, and we revised the paper to include more citations on prior work using multi-step objectives for training graph simulators. We highlight that our primary goal in this paper is the **PDE-matching setup** in a physical modelling setting with fixed-horizon trajectory-level supervision on the original graph with a port-Hamiltonian core, which avoids autoregressive drift and enables long horizons, hence improving long-range propagation. We revised the paper to highlight this point.
>
> The symplectic integrator (symplectic Euler) is applied for simplicity and stability, to ensure energy conservation of the Hamiltonian dynamics, and we analyze the energy behavior in the revision (App. I, Fig. 7).
>
> > Theorem 2 is almost identical to a sensitivity result in [2],…
> >
>
> We thank the reviewer for pointing out the connection of our work to [2]. Our sensitivity statement intentionally builds on the result of [2]. In general, we did not claim Theorem 2 as novel (as already stated in App. E.2), and in the revision, we now make this explicit in Section 4. We mainly extend the result in the IGNS setting, i.e., PDE matching for physical modelling. Furthermore, we extended the discussion (from line 343 in the revision) for the full port-Hamiltonian dynamics with damping/forcing used in our setting; here, the sensitivity bound does not collapse exponentially as in standard GCN depth. This is also supported empirically by the long-range behavior and energy analyzes in Fig. 6-7.
>
> > The core idea of the paper is closely related to the idea in [2]
> >
>
> We thank the reviewer for this point. We admit that we do take inspiration from [2] and use their port-Hamiltonian core, but the settings differ substantially. [2] targets static graph benchmarks (node/graph classification), while our work addresses **physical modeling on spatio-temporal graphs** with a focus on rollout stability over long horizons in both spatial and temporal domains.
>
> To address fairness, we added a [2]-style variant in Tab. 3(a): no multi-step supervision (final-step only). This variant underperforms in our setting because the dynamics between $0$ and $T$ are unconstrained, which is exactly where error compounds.
>
> We recall **our goal is to do PDE matching**: we integrate a port-Hamiltonian system and supervise the full trajectory at a fixed horizon (train/test), rather than training on short windows and rolling out autoregressively. This choice is justified by our universality result (Theorem 1) and extensively studied via experiments. Warmup (for global context at $t=1$) and geometric encoding (irregular meshes) are supporting pieces. In addition, we report a time-varying ablation for expressiveness, reported in Fig. 6.
>
> Overall, the revised experiments include the [2] baseline and clarify why PDE matching is necessary in our spatio-temporal regime.

---

> ### Author Response · Authors · 2025-11-21
> **Response to Reviewer eSDm - Part 2**
>
> > The ablation version of damping and forcing/residual terms is missing
> >
>
> We thank the reviewer for the suggestion. We now include the ablated model without damping and forcing (i.e., a conservative Hamiltonian core with a symplectic integrator and standard multi-step loss). Results are added in Table 3(b,c) and analyzed in Fig. 6-7. As shown, the conservative variant preserves energy (Fig. 7, right) but underperforms the full port-Hamiltonian variants with forcing and damping.
>
> These findings indicate that the non-conservative residual terms are important for accuracy and stability in our PDE-matching setup. As in our proof of Theorem 1, the external forcing term is needed, and serves as an input signal to the (universal) neural oscillator that can output any possible continuous signal. Furthermore, without this term, the model is limited to purely Hamiltonian systems or to dissipative ones, and does not have access to geometric information.
>
> > The other fundamental difference is use of symplectic integrator and warmup iteration…
> >
>
> We note that the ablation on the warmup was already present in Fig 3.b for the quantitative results.  For qualitative effect, see Fig. 12 with extended discussion in Appendix I. As noted in the main text, warmup mainly helps when global context is needed.
>
> We chose symplectic Euler for simplicity, as it is a common approach [2] that limits numerical dissipation and aligns with our energy-preserving core. While we agree that exploring other choices would be interesting, we believe this to be orthogonal to our contribution in PDE matching and out of scope for the present submission.
>
> > The paper addresses the long-range propagation problem
> >
>
> We thank the reviewer for the suggestion. Changing spatial resolution substantially alters the graph and signal distribution and, for GNN-based simulators, generally requires retraining. Multi-resolution generalization is mainly supported by spectral models (e.g., FNO). Our focus is therefore on tasks that are intrinsically long-range at a fixed resolution, where accurate simulation requires non-dissipative propagation across the entire domain.
>
> Our warmup ablation (Fig. 3) quantifies this need for wider receptive fields right at $t=1$, and the energy analysis (Fig. 7) shows the Hamiltonian core avoids dissipation during rollout, which is critical for long-range transport. We agree that a resolution study is complementary; since GNNs must be retrained per mesh, this is orthogonal to our claim (long-range difficulty stems from global propagation and non-dissipation, not only from node count).
>
> > How exactly does the warmup iteration work?
> >
>
> > Is the port-Hamiltonian used in the warmup phase shared with the time-evolving forward model?
> >
>
> We detailed the warmup step in Sec. 3.2 and Fig 1. In summary, we just iterate Eq. 8 for $L$ steps (i.e., the number of warmup steps) before running the actual simulation. Then we consider the final state of the warmup as the initial condition for the simulation. Hence, it is shared with the time-evolving forward.
>
> > Why is it fine to set  $D$ to be $0$ without the loss of generality
> >
>
> > What does B represent at line 989
> >
>
> In the setting of the universality theorem, we want to show that our model is general enough to learn any functional under certain assumptions, as discussed in the paper and in our previous comments. Our theorem shows that, without the term $D$, which is set to $0$ in our proof, our model can be cast more simply into the form of a Neural Oscillator (as in [1]), hence showing the Universality of IGNS. If we also allow for $D$ to be a general dissipation term, our proof still works (we just need to insert it into the definition of the auxiliary function), while our model gains flexibility to learn an even wider class of functionals. On the other hand, if required by the physical system, our model can simply learn a null dissipative term. We expanded the discussion on this matter in our proof.
>
> We thank the reviewer for pointing out the issue on the parameters $B$. Initially, in Theorem 3, we reported the original universality theorem of the Neural Oscillator by Lanthaler et al. [1], based on the form in Eq. 18 of our paper. However, we cast IGNS into the formulation in Eq. 19. Which is closer and still universal (as discussed in [1]). For clarity, we revised Theorem 3 to use the formulation in Eq. 19.
>
> > Typos
> >
>
> We fixed the mentioned typos and inaccuracies in the revision and made sure to check the remaining paper for clarity and consistency.
>
> [1] Neural Oscillators are Universal, NeurIPS 2023
>
> [2] Port-Hamiltonian Architectural Bias for Long-Range Propagation in Deep Graph Networks, ICLR 2025

---

> ### Comment · Reviewer_eSDm · 2025-11-25
>
> Thank the authors for the rebuttal and for addressing my concerns. I have follow-up comments and questions for the rebuttal.
>
> Unlike the authors’ view, I am still not convinced to see Theorem 2 as an extension. I agree that the settings and applications of the present work differ from those of [2]. However, I would not count this difference as a factor that makes the theorem seem an extension of a theoretical argument; instead, see it as a mere difference in instances of graph systems to which the essentially identical theoretical arguments are applied. It also seems like circling back to the original port-Hamiltonian theory that inspired and was adapted in [2]. As the authors also noted, the paragraph starting at line 343 makes an effort to provide insight into the sensitivity of the general form of the proposed Hamiltonian. However, most of the statements are the elucidation of GNN-CON, and its link to the proposed idea and the indication remain unclear. I believe this stems from unaccounted mapping the authors have in mind between the formulations of GNN-CON and the paper. Perhaps, applying proofs from some of the results in GNN-CON to the port-Hamiltonian formulation could yield new insights and/or intriguing results, since GNN-CON assumes different MPNN architectures from the one in the submission.
>
>
> Although I am still uncertain about how the warmup phase exactly works, if the model used in the time-evolving forward phase is shared during the warmup iteration, does that not break the physics? The validity of the proposed integration framework is not apparent to me, because the warmup will implicitly embed information obtained through time evolution into the output, and when $t < L$, a simulation from this output can be seen as an evolution backwards in time. Do the authors have a justification for the warmup preserving the temporal order and, more broadly, respecting the physics laws underlying the physics system?

---

> > ### Author Response · Authors · 2025-11-27
> > **Response to the follow-up questions**
> >
> > We thank the reviewer for the follow-up questions.
> >
> > > Discussion about GraphCON
> > >
> >
> > Firstly, to better explain the text around line 343, we extended the discussion and added Appendix E.3. There, as suggested, we follow the structure of Prop. 3.5/3.6 in GraphCON [1] and calculate the gradient of the loss w.r.t. the parameters. Concretely, we start from the symplectic-Euler discretization of Eq. 8, and calculate the one-step Jacobian $\frac{\partial x^{t+1}}{\partial x^{t}}$ and then apply the chain-rule to the remaining steps to obtain the $\frac{\partial x^{T}}{\partial x^{s}}$ similar to what has been done in GraphCON. In the end, we arrived at a similar form, where the final term $\frac{\mathcal{L}^T}{\partial \mathbf{w}^{(t)}}$  has the leading order of $O(\Delta t^2)$, which is independent of the number of steps/layers, i.e., no exponential decay with depth.
> >
> > > Warm-up phase
> > >
> >
> > Secondly, to clarify the warmup phase, we first describe one forward pass. Starting from the encoded state $x_0$, we apply the symplectic-Euler update of Eq. (6) repeatedly to obtain $x_1, x_2, \ldots, x_{L+T-1}$. Here, **warmup phase is just the phase before we start to decode and compute the loss.**
> >
> > Without warmup, we decode at $t=1$ and supervise $[1,\ldots,T]$; with warmup $L$, we decode at $t=L$ and supervise $[L,\ldots,L+T-1]$. The same update (shared weights) is used at every step. There is no backward evolution and no use of future information in the forward pass, warmup is simply a burn-in of the same forward integrator. Because each step is exactly the Eq. (6) update of the port-Hamiltonian system, temporal order is respected and the underlying physics remains intact (up to standard discretization error).
> >
> > We hope these clarifications address your concerns. We’re happy to follow your guidance on any remaining points and, given the added derivation (Appendix E.3), the clarified warmup explanation, and the new analyses, we kindly ask you to consider revisiting your score.
> >
> > [1] Graph-Coupled Oscillator Networks, Rusch et. al. ICML 2022

---

### Official Review · Reviewer_KM4o · 2025-10-29

**Soundness:** 3
**Presentation:** 4
**Contribution:** 3
**Rating:** 6
**Confidence:** 3

**Summary:**

This paper presents Information-preserving graph neural simulators (IGNS). It is a novel approach aimed to improve long-range propagation of information and to reduce error accumulation in task related with physical modelling. By using port-Hamiltonian dynamics, IGNS preserves information across graphs and captures conservative and non-conservative dynamics. Authors use some key innovations, for example: a warmup phase for initializing global context, geometric encoding for irregular meshes, and a multi-step training objective for stable long-horizon predictions.

Additionally, the authors provide strong theoretical guarantees for IGNS's universality and gradient preservation. Authors provide comprehensive experiments on six benchmarks, including new tasks. IGNS consistently outperforms many baselines.

**Strengths:**

1. The idea is novel and interesting.
2. The paper has solid theoretical foundations (universality, non-vanishing gradients) and robust experimental design against many strong baselines across different physical systems.
3. The paper is generally well-written, logically structured, with clear problem statements and architecture descriptions. Good use of figures, tables, and appendices.
4. It offers a substantial advancement in GNSs by solving critical long-standing limitations.

**Weaknesses:**

1. A direct ablation for authors' static geometric encoding feels like missed. The authors make an argument that model's specific geometric encoding helps avoid "overfitting," which is important for generalizing well. However, authors don't really show us an experiment where they directly compare their own model with and without developed static encoding. Instead, they rely an indirect comparison to other architectures.
2. No inference time analysis. It it is hard to understand, beyond just training time, how fast do the models predict one or multiple steps during inference? This is crucial for "accelerate simulations".
3. No scale analysis. How does the computational cost scale with increasing graph size (nodes, edges) or rollout horizon?

**Questions:**

1. See W2 and W3. It is interesting to see results of inference time and scale analysis.

2. In the message-passing update you use only q_{i} and q_{j} (see eq. 9). Does this exclude the momentum variables p_{i} and p_{j} from exchange between neighbors, or is q there meant to denote the full node state [q,p]? Please clarify.

3. Did you conduct an ablation study to directly compare static geometric encoding of IGNS against a dynamic geometric encoding strategy (e.g., updating edge features per time step) within the IGNS framework (not changing the model)? It is needed to empirically validate its claimed benefit in preventing overfitting. If so, where are these results presented? If not, please conduct this ablation study.

---

> ### Author Response · Authors · 2025-11-21
> **Response to Reviewer KM4o**
>
> We thank the reviewer for their time and effort and for acknowledging the novelty in the idea, and the solid theoretical foundations as well as the clear presentation . Below, we provide answers to the reviewer's concerns.
>
> > A direct ablation for authors' static geometric encoding feels like missed….
> >
>
> We recall that the claim we made mainly shows that rewiring methods can make the message heavily dependent on the geometries (e.g. MGN-rewiring), as detailed in Tab. 2(a) and the qualitative results in Fig. 5. To address the request for a direct ablation of the geometric encoding, we now compare IGNS with vs. without our static geometric encoding on a new SphereCloth-direct task. As shown in Fig. 6(a) and Tab. 3(a), the model still converges without geometric encoding but is less accurate; adding the encoding improves performance. This supports our design choice: unlike rewiring (which can overfit to added edges), our method does not inject geometry into the message kernel and uses geometry only in the external forcing, reducing geometry-specific overfitting while still benefiting from geometry if necessary.
>
> > No inference time analysis…
> >
>
> > No scale analysis….
> >
>
> As also pointed out by Reviewer S8R7, we added a complexity analysis and runtime study in App. H. We refer to the general response for a short summary of this analysis.
>
> > In the message-passing update you use only q_{i} and q_{j} (see eq. 9)…
> >
>
> Eq. (9) shows the edge message built from $(q_i,q_j)$ because the geometric encoding is defined on coordinates and is computed only from $q$, not $p$. Nevertheless, this does not exclude momentum from the dynamics, as shown in Eq. 8. In the end, each message step uses the geometrical information from $q$ and consumes $p$ via Eq. 8 to let the information flow through the full state $[q, p]$.
>
> > ablation study to directly compare static geometric encoding of IGNS against a dynamic geometric encoding strategy
> >
>
> Thank you for the suggestion. However, we recall our overfitting claim specifically targets direct rewiring methods. In contrast, IGNS does not inject geometry into the message kernel and uses geometry only in the external forcing, which we additionally ablate in Fig. 6.

---

> > ### Comment · Reviewer_KM4o · 2025-11-27
> > **Follow-up**
> >
> > Thank you. Overall, I am satisfied with experiments. As for now, the paper provides an analysis of the computational complexity of IGNS, which is stated to scale linearly with the number of nodes and edges in the graph. Could you support this theoretical analysis by additional empirical scaling analysis where you vary the number of points (and edges, of course) in the simulation grid and report computational costs? This could further strengthen the claims made in the paper.

---

> > > ### Author Response · Authors · 2025-11-27
> > > **Response to the follow-up question**
> > >
> > > Thank you for your follow-up question and for noting you are satisfied with the experiments. As suggested, we added an **empirical scaling analysis** (same setup as in our previous analysis) and included the plot in **Figure 5**; we also provide the table below for easy reference.
> > >
> > > **IGNS (ours)**
> > >
> > > | Resolution | Inference time (s) |
> > > | --- | --- |
> > > | 24×24 | 0.150400 ± 0.002432 |
> > > | 48×48 | 0.148725 ± 0.000741 |
> > > | 96×96 | 0.149625 ± 0.000591 |
> > > | 192×192 | 0.279475 ± 0.019647 |
> > >
> > > **MGN**
> > >
> > > | Resolution | Inference time (s) |
> > > | --- | --- |
> > > | 24×24 | 0.622575 ± 0.003296 |
> > > | 48×48 | 0.645300 ± 0.002486 |
> > > | 96×96 | 1.624375 ± 0.228426 |
> > > | 192×192 | 7.277450 ± 0.811855 |
> > >
> > > As shown, IGNS remains near-flat up to 96×96 and grows moderately at 192×192 (consistent with linear scaling). By contrast, MGN grows sharply with resolution because it performs **sequential spatial updates** ($S=15$ in this case) at every step, which under-utilizes the GPU and scales poorly with graph size.
> > >
> > > We’re happy to follow any further guidance you may have. Given your satisfaction with the experiments and these added results, we kindly ask you to consider raising your score.

---

### Official Review · Reviewer_Rdux · 2025-11-01

**Soundness:** 3
**Presentation:** 1
**Contribution:** 2
**Rating:** 6
**Confidence:** 4

**Summary:**

This paper introduces Information-preserving Graph Neural Simulators (IGNS) for learning physical dynamics of  both conservative and non-conservative port-Hamiltonian systems. Key features of the work are: 1) Hamiltonian formalism to eliminate part of the model systematic drifts; 2) a warmup phase for global context initialization; 3) a multi-step training loss for long-horizon stability.
The authors also present new benchmarks (Plate Deformation, Sphere Cloth, Wave Balls) to test long-range dependencies.
IGNS achieves good results across all datasets, showing higher or comparable accuracy and stability than MeshGraphNets and GraphCONs together with some other methods.

**Strengths:**

* novelty in introducing a port-Hamiltonian formalism to graph simulators
* theoretical proofs of universality and non-vanishing gradients
* data efficiency, importance of warm-up steps and length of confident prediction horizon were investigated
* code attached

**Weaknesses:**

* the advantages of Hamiltonian dynamics simulation were not properly studied (like the energy conservation for the conservative systems)
* the generalizability is under question
* the file dataset.zip in supplimentary link is corrupt
* the paper lacks qualitative discussion of the distinction between two versions of the algorithm is not clear (IGNS, IGNS_ti (time-independent))

**Questions:**

* What are the computational expensies of your model training and inference compared to competitors and numerical simulator?
* What are the limits of your model generalization? (out of distribution, new geometries, etc.)
* Why for Kuramoto-Sivashinsky equation the results for IGNS_ti are so much better than for IGNS?

---

> ### Author Response · Authors · 2025-11-21
> **Response to Reviewer Rdux**
>
> We thank the reviewer for their time and effort and for acknowledging the novelty in introducing the port-Hamiltonian to graph simulators. Below, we provide answers to the reviewer's concerns.
>
> > • the advantages of Hamiltonian dynamics simulation were not properly studied (like the energy conservation for the conservative systems)
> >
>
> We now added an **energy conservation analysis** in App. I. In Fig. 7, the right panel (no forcing and no damping) shows a conservative Hamiltonian system: normalized energy oscillates and, with smaller step sizes, stays nearly constant. The left panel (full port-Hamiltonian with forcing and damping) shows energy increase due to learned non-conservative effects. This study confirms the expected energy behavior.
>
> > the generalizability is under question
> >
>
> > What are the limits of your model generalization? (out of distribution, new geometries, etc.)
> >
>
> **Different geometries.** This is the main reason we use graph networks. IGNS mainly operates at the node level and does not assume a fixed mesh. In our experiments (WaveBalls, CylinderFlow, ImpactPlate), the geometry varies across train/val/test, and IGNS generalizes well. In contrast, rewiring methods can overfit to the extra edges they add; we observe this behavior on the Plate Deformation task when the test geometry (with unseen force nodes) differs from the training (see Tab. 2a). In conclusion, IGNS does not rely on fixed rewiring and is more robust to geometric changes within the training distribution.
>
> **Out-of-distribution limits.** Like other standard GNN simulators, IGNS does not zero-shot generalize to different resolution changes without retraining. Local message passing learned to set an effective propagation speed and radius; changing resolution at test time changes these scales and degrades accuracy. The same caveat holds for strong shifts in boundary conditions, forcing statistics, or material parameters not seen during training.
>
> > the file dataset.zip in supplimentary link is corrupt
> >
>
> We thank the reviewer for pointing us to this problem. While we did not encounter any issues, we suggest downloading the dataset.zip file and opening it with any zip decompression tool. If it still does not work, we are happy to re-upload it.
>
> > the paper lacks qualitative discussion of the distinction between two versions of the algorithm is not clear (IGNS, IGNS_ti (time-independent))
> >
>
> The original submission included a brief qualitative comparison on SphereCloth in what is now Fig. 8.  IGNS produces a smoother, more coherent surface, while the IGNS_ti surface appears rougher. In the revision, we also added WaveBall $T=200$, where the visualization is clearer. Here, IGNS shows that it can capture high-frequency waves, whereas IGNS_ti blurs peaks and loses detail (see the files in folder rebuttal_wave_ball_200 in the Supplement)
>
> Furthermore, we add another ablation on Tab. 3 (b, c) and the learning curves in Fig. 6 to analyze the role of damping and forcing of IGNS and IGNS_ti. Here, the time-varying IGNS can still reach a viable solution without forcing/damping (after longer training), but the time-independent IGNS_ti struggles, resulting in a shallow slope in the validation curves. This clarifies the benefit of time-varying weights.
>
> > What are the computational expensies of your model training and inference compared to competitors and numerical simulator?
> >
>
> As also pointed out by Reviewer S8R7, we added a complexity analysis and runtime study in App. H. We refer to the general response for a short summary of this analysis.
>
> > Why for Kuramoto-Sivashinsky equation the results for IGNS_ti are so much better than for IGNS?
> >
>
> In KS, we reported the test MSE ($\times 10^{-3}$), meaning that the results of IGNS and IGNS_ti after scaling are quite close, i.e., both learn the KS dynamics. We recall that, the provided supplement already includes visualizations to inspect this gap (Fig. 8: KS-1 and Fig. 9: KS-2). Typically, KS shows chaotic behavior in the later stage, while the early stage is more stable with smaller density changes. This is reflected in the error scales in Fig. 10 (early) vs. Fig. 11 (late) in the revised paper. We hypothesize that IGNS, with its time-varying core, tends to capture high-frequency structures in the later stage rather than over-fitting fine details in the early stage, which explains the small residual gap. Importantly, both IGNS variants succeed here, while FNO-RNN (misses local detail) and MGN (error accumulation) struggle.

---

> ### Author Response · Authors · 2025-11-27
> **Follow-up on Review Feedback**
>
> Thank you again for your thoughtful review and constructive suggestions. We’re following up since we haven’t seen a response to our rebuttal yet, and we’d be happy to discuss further before the discussion phase ends.
>
> To briefly summarize, we provided:
>
> - An energy conservation analysis (App. I, Fig. 7) showing conservative vs. full port-Hamiltonian behavior, plus a validation-curve(Fig. 6).
> - A clarification that IGNS, like other graph-based neural simulators, generalizes across geometry variation within distribution.
> - IGNS vs. IGNS_ti: added qualitative comparisons (SphereCloth; WaveBall-200): IGNS shows that it can capture high-frequency detail than IGNS_ti.
> - A complexity analysis + runtime table: IGNS shows that it is faster than MGN.
>
> We hope these additions address your concerns. We’re happy to follow your guidance on any remaining points and would kindly ask you to consider revisiting your score.

---

### Author Response · Authors · 2025-11-21
**Answer to all reviewers**

We thank the reviewers for acknowledging the clear presentation, the comprehensive empirical study, and the provided theoretical justification of our work.

We see three common concerns among the reviews, namely method clarity, ablations on core components, and complexity analysis. To address these and individual concerns, we revised the submission, with changes marked in Blue. In particular, we want to highlight:

1. **Method clarity**. Our goal is to improve long-range dependencies in Graph Neural Simulators. To this end, we use a port-Hamiltonian system as the dynamics core. Our Theorem 1 shows that integrating this system can approximate the target dynamics. This enables **PDE matching**, where we can **supervise the full trajectory with a fixed-window** $T$ **using multi-step loss at both training and testing**. This process mirrors classical PDE solvers, where information propagates in space as time advances. Empirically, we show that we can obtain high accuracy on complex tasks like Wave Balls with long horizons (up to $T=200$), where autoregressive methods such as MGN fail severely (see Tab. 12).

    We recall that prior work typically uses short $T \leq 20$ in both training and testing [1,2] or trains on short windows and then rolls out autoregressively [3], or applies multi-step supervision only in a reduced latent space [4] rather than on the original graph.

2. **Ablations and analysis**. We added an energy conservation analysis (App. I). In Fig. 7 (in the new revision), the conservative core (no forcing/damping) keeps normalized energy oscillating or nearly constant with sufficiently small $\Delta t$. Meanwhile, the full model exchanges energy as expected. In addition, Fig. 6 investigates the contribution of individual terms, showing validation curves for IGNS/IGNS_ti without forcing and/or damping, as well as without the warmup phase or geometric encoding. This ablation explains why the full model IGNS with all components obtains the highest accuracy.
3. **Complexity and runtime.** IGNS keeps the graph convolution backbone’s linear cost w.r.t. node and edge count, meaning one step is $\mathcal{O}(∣V∣+∣E∣)$; Assuming $L$ warmup steps and rollouts of length $T$, the total cost is, the total cost is $\mathcal{O}((L+T)(∣V∣+∣E∣))$. For comparison, MGN needs $S$ message-passing updates per step (often $S=15$), resulting in a higher cost of $\mathcal{O}(ST(∣V∣+∣E∣))$. We added an inference runtime table in Tab. 8, which consistently shows that IGNS is faster than MGN.
4. **Two new task variants.** The additional ablations were done on two new task variants, namely, *SphereCloth-direct* (where we only connect the ball to four corners) to probe long-range propagation and the contribution of individual components, and *Wave Balls-200* (trained with curriculum scheduling as in [1]) to test long-horizon supervision capability and run-time analysis.

Again, we thank the reviewers for their efforts and provide detailed answers to their individual reviews to clarify and address their individual concerns. We hope our clarifications and revisions will address your concerns and prompt a reconsideration of your evaluation and score.

[1] Learning the Dynamics of Physical Systems from Sparse Observations with Finite Element Networks, ICLR 2023

[2] RoboCook: Long-Horizon Elasto-Plastic Object Manipulation with Diverse Tools. CoRL 2023

[3] Eagle: Large-Scale Learning of Turbulent Fluid Dynamics with Mesh Transformers. ICLR 2023

[4] Predicting Physics in Mesh-reduced Space with Temporal Attention. ICLR 2022

---

### Author Response · Authors · 2025-11-28
**Post-data leak: Summary of rebuttal updates**

We thank the organizers for their transparency regarding the data leak. Given the decision to revert reviews and scores to their pre-rebuttal state, we provide this short summary for the new Area Chairs.

During the discussion, we closely followed the reviewers’ feedback and addressed their concerns. Reviewer **KM4o** confirmed satisfaction with the new experiments and asked for an additional empirical scaling analysis, which we subsequently added. Reviewer **eSDm** replied with two follow-up questions. We then extended the theory discussion and posted clarifications to those questions accordingly. Unfortunately, **S8R7** and **Rdux** did not respond to our rebuttal despite our best efforts to address their concerns, and the abrupt close of the discussion prevented further dialogue.

We respectfully ask the new Area Chairs to consider our comprehensive responses and the improvements made during the rebuttal in their final decision.

Kind regards,

The Authors

---

### Meta-Review · Area_Chair_As3c · 2026-01-06

**Summary:**

The reviewers raised the following main concerns:

1. (Reviewer Rdux): The advantages of Hamiltonian dynamics simulation were not properly studied.

2. Complexity and runtime (by reviewers Rdux, KM4o, S8R7).

3. Scale analysis: How does the computational cost scale with increasing graph size (nodes, edges) or rollout horizon? (by reviewers KM4o)

4. Reviewer eSDm commented that the core idea of the paper is very similar to [1], where relevant comparison and ablations are missing. This is an important criticism.

5. Reviewer eSDm commented on the overclaim and inaccurate description of Theorem 1.

[1] Heilig, et.al., "Port Hamiltonian Architectural Bias for Long-Range Propagation in Deep Graph Networks.", ICLR 2025.

**Reviewer Concerns:**

I read the revised paper and the rebuttal. I think most of the concerns by the reviewers are resolved. After comparing with the paper [1], I think the difference of the current paper to [1] is significant enough for a presentation at the conference, which also brings novel ideas to the neural PDE community.

**Reviewer Scores:**

For reviewers Rdux, KM4o, I think they will remain their score of 6. Reviwer S8R7 may keep the score of 4 or raise to 6. Reviewer eSDm may raise the score to 4.

---

### Decision · Program_Chairs · 2026-01-26

Accept (Poster)